# Circuit asymmetries underlie functional lateralization in the mouse auditory cortex

Robert B. Levy[1], Tiemo Marquarding[1,5], Ashlan P. Reid[1], Christopher M. Pun[2], Nicolas Renier [3] & Hysell V. Oviedo [1,4]

The left hemisphere's dominance in processing social communication has been known for over a century, but the mechanisms underlying this lateralized cortical function are poorly understood. Here, we compare the structure, function, and development of each auditory cortex (ACx) in the mouse to look for specializations that may underlie lateralization. Using Fos brain volume imaging, we found greater activation in the left ACx in response to vocalizations, while the right ACx responded more to frequency sweeps. In vivo recordings identified hemispheric differences in spectrotemporal selectivity, reinforcing their functional differences. We then compared the synaptic connectivity within each hemisphere and discovered lateralized circuit-motifs that are hearing experience-dependent. Our results suggest a specialist role for the left ACx, focused on facilitating the detection of specific vocalization features, while the right ACx is a generalist with the ability to integrate spectrotemporal features more broadly.

[1] Biology Department, The City College of New York, New York, NY 10031, USA. [2] The City College of New York, Macaulay Honors College, New York, NY 10031, USA. [3] Institut du Cerveau et de la Moelle Epinière, Paris 75013, France. [4] CUNY Graduate Center, New York, NY 10016, USA. [5] Present address: Institute for Molecular and Cellular Cognition, Center for Molecular Neurobiology Hamburg, University Medical Center Hamburg-Eppendorf, Hamburg 20251, Germany. Correspondence and requests for materials should be addressed to H.V.O. (email: hoviedo@ccny.cuny.edu)

More than 150 years ago, Paul Broca reported the left hemisphere's language dominance[1], launching a still unresolved puzzle in neuroscience: the origin, function, and mechanisms of lateralized auditory processing. Since then, behavioral and imaging studies in humans have confirmed this functional asymmetry[2]. The leading hypothesis concerning its teleology is that division of labor between the hemispheres can improve cognitive performance: the efficiency of decoding multiple features from rapidly changing sounds is improved by lateralization[3,4]. Functionally, the left auditory cortex (ACx) is postulated to specialize in fast semantic processes (identifying specific sequences in speech), and the right ACx in prosody (pitch and intonation). Therefore, each hemisphere may specialize in processing different spectrotemporal features of sounds, but the mechanisms remain unresolved. Because lateralization is evolutionarily conserved across many vertebrates, including mice[5], we took advantage of the powerful analysis tools available in animal models to determine the underlying neural mechanisms. Identifying the mechanisms of lateralized functions is important not only for a basic understanding of the nervous system but also for mental health. Abnormal lateralization of language processing in the human brain has been identified as a biological risk factor for schizophrenia[6] and is linked to autism[7,8].

There is behavioral evidence that mice process intraspecies vocalizations preferentially in the left ACx. Studies in lactating dams and virgins trained to retrieve pups have implicated the left ACx in the processing of pup-calls[9,10], whereas other studies have implicated the right ACx in more general auditory computational tasks like the detection of sweep direction[11,12]. The behavioral evidence for lateralized auditory processing in the mouse together with the unique circuit features we discovered previously in the ACx[13], led us to investigate mechanisms of lateralized auditory function. Here, we report lateralized activation of the left ACx and right ACx when mouse vocalizations and frequency sweeps are presented during passive listening. To dissect the mechanisms underlying lateralized auditory processing we compared the spectrotemporal selectivity, synaptic connectivity, axonal projections, development, and experience-dependence of the left and right ACx. We found hemispheric differences from circuits to function that provide a comprehensive framework of the underlying division of labor for processing complex sounds.

## Results

**Vocalizations and sweeps lateralize activation in the ACx.** To screen for neuronal populations with lateralized activation in the mouse ACx, we played vocalizations and frequency sweeps (henceforth sweeps) to freely moving adult, male CBA mice in a sound booth. Measuring the brain-wide activity of individual neurons in real-time is challenging; therefore, we used expression of the immediate early gene Fos as a marker of neural activity. We combined immunolabeling with the iDISCO brain-clearing technique and quantitative volume analysis, which provides an unbiased analysis of whole-brain detection of Fos[14,15]. Mice were placed in a sound booth individually and presented with either sweeps or vocalizations. Their brains were subsequently processed with iDISCO and immunolabelled to detect Fos activity. To quantify Fos-positive neurons, we performed 3D volume imaging and used ClearMap to analyze the image stacks and perform statistical significance tests (see Methods). The group of mice presented with sweeps had significantly more Fos-positive neurons in the right ACx than the left ACx (Fig. 1a, b, compare top panels, $p = 0.05$, $n = 3$ mice). A coronal projection of the 3D volume data revealed that the highest activation was largely in superficial cortical layers in the primary ACx (Fig. 1c, d, top). A parasagittal cortical projection of the 3D volume data also showed

that sweeps led to activation throughout the primary ACx in the right hemisphere (Fig. 1e–h top). In contrast, mice presented with vocalizations had significantly more Fos-positive neurons in the left ACx than the right ACx (Fig. 1a, b, compare bottom panels, $p = 0.05$, $n = 3$ mice). Similar to the activation of the right ACx by sweeps, vocalizations also produced the strongest activity in superficial layers of the left ACx (Fig. 1c, d, bottom). Contrary to the broad activation of the right ACx by sweeps, vocalizations led to activation in anterior portions of the primary ACx (Fig. 1e–h, bottom), where ultrasonic frequencies are represented[16]. To examine whether lateralized activation by these complex sounds first arises in the ACx, we analyzed the distribution of Fos-positive neurons in the auditory thalamus (Fig. 1i–k). Neither sweeps nor vocalizations led to significantly lateralized activation in the medial geniculate nucleus (Fig. 1k). These data are the first to show a direct comparison in the activation of the left and right ACx in the same animal with single-cell resolution. Together these data extend and confirm previous behavioral and pharmacological studies showing that sounds with different spectrotemporal features differentially activate the ACx. Our results also suggest that intracortical mechanisms in superficial layers of the left and right ACx are directly engaged in the division of labor in auditory processing.

**Sweep direction selectivity is lateralized in the ACx.** To screen for spectrotemporal selectivity differences between the auditory cortices, we performed cell-attached recordings in anesthetized CBA male mice ($n = 23$ mice). For this survey, we varied the speed and direction of sweeps, because these are basic components of dynamic stimuli and easier to parametrize than vocalizations. As the hemispheric differences in Fos activation largely arose in superficial layers, we targeted neurons located < 300 μm from the pial surface. In a previous study of the rat's right ACx, it was reported that sweep direction selectivity was parallel to the tonotopic axis. Neurons with low-frequency tuning preferred rising continuous sweeps (up sweeps), whereas neurons with high-frequency tuning preferred falling continuous sweeps (down sweeps[12]). Therefore, we expected to find similar trends in the left and right ACx of the mouse. In the left ACx we found neurons with a variety of best frequency (BF) tuning (Fig. 2a, b left panels); but interestingly, regardless of a neuron's BF there was a prevalent preference for down sweeps (Fig. 2a, b middle and right panels). On the other hand, in the right ACx we found diverse BF tuning (Fig. 2c, d left panels), but sweep direction preference depended on BF (Fig. 2c, d middle and right panels), similar to what was reported in the rat. To quantify these observations, we first calculated the sweep direction selectivity index (DSI) for each cell at its preferred sweep speed for each direction (see Methods). We found that the distribution of sweep direction selectivity was indeed skewed to down sweeps in the left ACx (Fig. 2e top, skewness = −0.74), but not in the right ACx where the distribution was symmetrical (Fig. 2f top, skewness = 0.16). Furthermore, the BF-DSI dependence also differed between the hemispheres. In the left ACx, there was no significant correlation between BF and direction selectivity (Fig. 2e bottom, Spearman $r = -0.15$, $p = 0.53$, $n = 19$ cells), whereas in the right ACx there was a significant dependence of direction selectivity on BF (Fig. 2f bottom, Spearman $r = -0.91$, $p = 7.1e-6$, $n = 14$ cells). The observation that neurons in the right ACx respond to any sweep direction (compared with the more-selective left ACx) supports the Fos activation data: relatively higher activity in the right ACx in mice presented with sweeps. We next examined the circuit mechanisms that could underlie the specialist and generalist response properties of the left and right ACx.

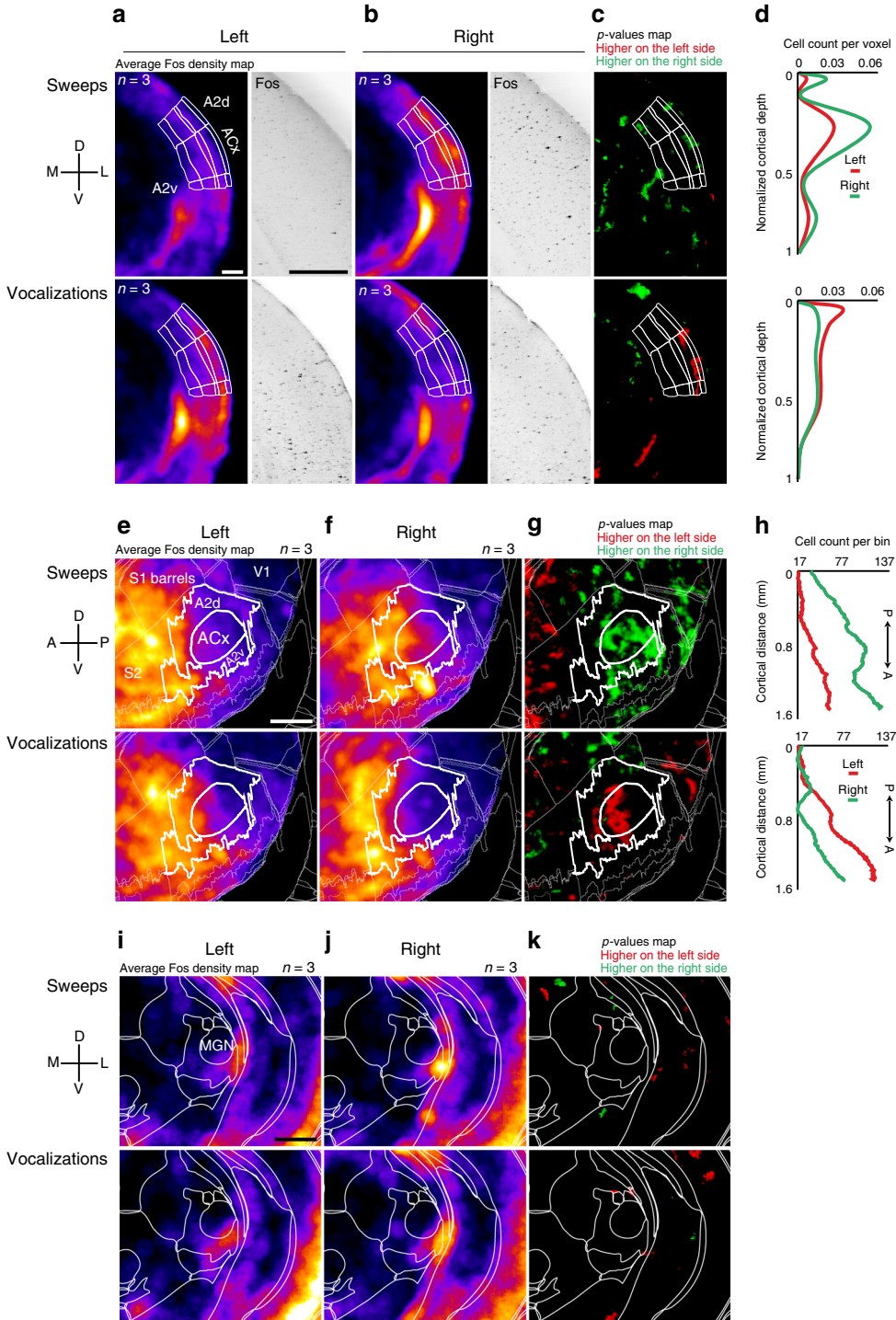

**Fig. 1** Sweeps and mouse vocalizations evoke lateralized activation in the ACx. **a** Coronal plane showing average Fos density (left) and Fos-positive cells stack (right) for sweeps (top, $n = 3$ mice) and vocalizations (bottom, $n = 3$ mice) in the left ACx; scale bar 0.5 mm. **b** Same as (**a**) but showing the right ACx. **c** Maps of statistically significant differences in activation between the left and right hemispheres. Sweeps evoked higher activation in the right ACx (top), and vocalizations evoked higher activation in the left (bottom). **d** Laminar distribution of cell count per imaging voxel ($25 \, \mu m^3$). **e, f** Lateral cortical projection showing average Fos density for sweeps (top, $n = 3$ mice) and vocalizations (bottom, $n = 3$ mice) in the left and right hemispheres (respectively); scale bar 1 mm. **g** Maps of statistically significant differences in activation between the left and right cortical projections. Sweeps evoked higher activation in the right ACx (top), and vocalizations evoked higher activation in the left (bottom). **h** Distribution of cell count along the tonotopic axis. The $y$ axis represents the number of cells in superficial layers in neighborhoods of 190 $\mu m$ around each point on the line drawn. **i, j** Coronal plane showing average Fos density in the auditory thalamus (MGN), for sweeps (top, $n = 3$ mice) and vocalizations (bottom, $n = 3$ mice) in the left and right hemispheres (respectively); scale bar 1 mm. **k** Statistical maps showing no significant difference in activation between the left and right MGN projections

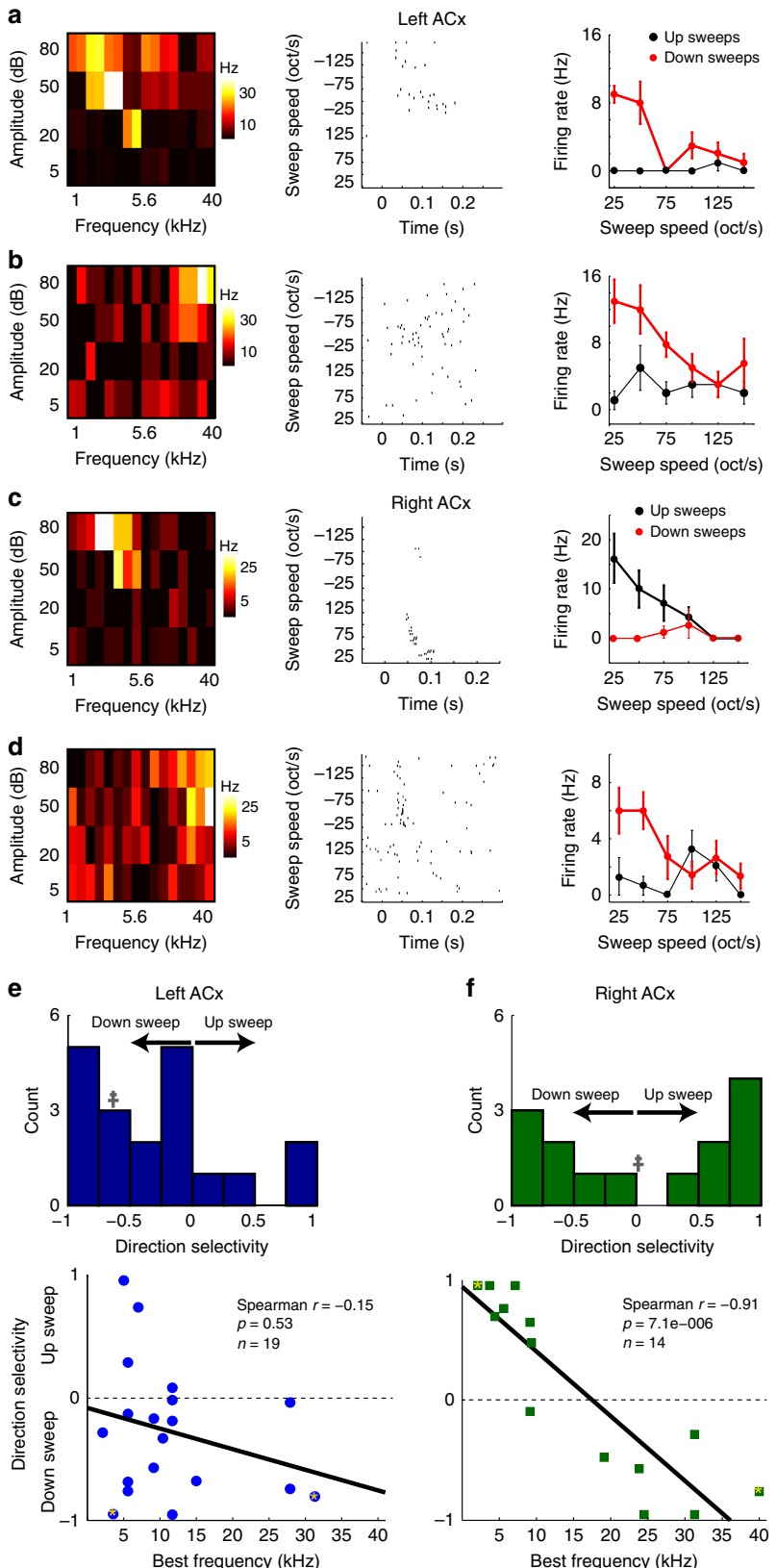

**Lateralized circuit-motifs in the ACx**. To screen for potential circuit mechanisms that could underlie the lateralized activation and functional differences found in vivo, we performed laser scanning photostimulation (LSPS) in slices of the left and right ACx. We targeted layer 3 (L3) pyramids for connectivity mapping because the Fos activation data indicated an important role of superficial layers in lateralized function. Furthermore, in a previous study we found asymmetries in pathways projecting to L3[13]. We performed voltage-clamp recordings in L3 pyramids along the tonotopic axis of the ACx (approximately 1 mm; Fig. 3a). Representative single-cell input maps recorded at different locations of the tonotopic axis in the left ACx show that the L6→L3

**Fig. 2** Hemispheric differences in sweep direction selectivity. Cell-attached recordings showing tone and sweep selectivity in L2/3 neurons. **a** Frequency response area (FRA, left), spike rasters of responses to sweeps (middle), and sweep selectivity (right) of a single neuron in the left ACx. The neuron's BF = 3.4 kHz and had a down-sweep preference. **b** Left ACx neuron with BF = 31.3 kHz and a down-sweep preference. **c** Right ACx neuron with BF = 3.4 kHz and an up-sweep preference. **d** Right ACx neuron with BF = 40 kHz and a down-sweep preference. All data are presented as mean ± SEM. **e**, **f** Distribution of sweep direction selectivity (top panels), and direction selectivity-best frequency dependence (bottom panels) in the left and right ACx. Double cross in the top panels represents the skewness of the distribution. Points with asterisks correspond to the cells shown **a**–**d**, and solid black lines are best linear fits

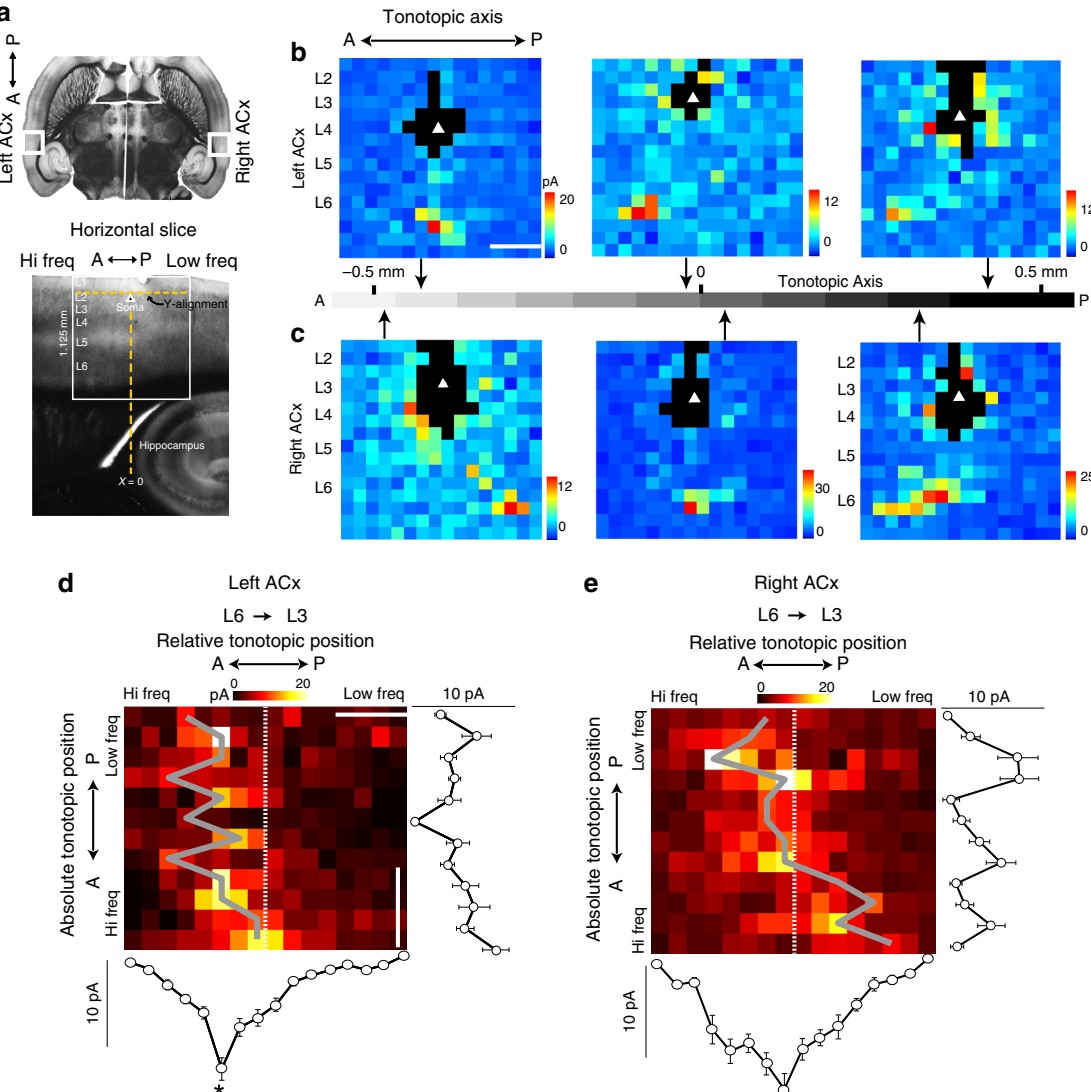

**Fig. 3** Tonotopic connectivity differs between the left and right ACx. **a** Horizontal slices capture the anterior–posterior (A↔P) representation of tonotopy in the left and right ACx (top); white box scale 1 mm × 1 mm. DIC image of a horizontal slice during an experiment depicting the anatomical landmarks used to align consistently the stimulation grid (bottom). **b**, **c** Left, middle, and right panels are representative single-cell maps of six L3 neurons mapped at anterior, middle, and posterior locations of the left and right ACx, respectively. Each cell is shown with its native position along the tonotopic axis (arrows); scale bar 0.3 mm. **d**, **e** 2D population maps where each row is the average L6 input to L3 cells mapped within 50 μm from one another and ordered by tonotopic position in the left ACx (n = 38) and the right ACx (n = 40), respectively. The x axis is the position of the cells in the LSPS map grid (cells are always centered on their input maps). The tonotopic position of each row is the average distance of the cells binned to the tip of the hippocampus. Dashed white line represents the soma position and gray lines connect pixels of maximum input strength. Line plots in **d** and **e** are the columnar and row averages of synaptic input calculated from the 2D population maps. Asterisk indicates that synaptic input arising from the anterior and posterior areas of the 2D population map were significantly different (p = 1.4e-3, n = 38, Wilcoxon rank-sum). X scale bar 0.3 mm, y scale bar 0.33 mm. All data are presented as mean ± SEM

pathway arose from L6 neurons located several hundred microns anterior to the L3 postsynaptic neuron (henceforth, out-of-column) throughout most of the ACx, and consistently from putative higher frequency bands (Fig. 3b middle and right). The L6

input became columnar at more anterior portions of the left ACx where the highest frequencies are represented (Fig. 3b left). Mapping the corresponding anatomical area in the right ACx revealed strikingly different connectivity from that found in the

left ACx. In anterior portions of the right ACx, L3 received out-of-column L5/6 input that was biased to lower frequencies (Fig. 3c left), middle sites received columnar input (Fig. 3c middle) and posterior sites received higher frequency biased L6 input similar to the left ACx (Fig. 3c right; raw synaptic input traces in Supplementary Fig. 1). This suggests the spectral integration scheme in L3 differs between the auditory cortices.

To examine the relationship between tonotopy and layer-specific input in our population, we constructed two-dimensional (2D) tonotopic input matrices by vectorizing layer-specific input and combining the input maps of L3 neurons in 50 μm bins (see Methods). If the input from a particular layer is columnar, then hotspots will be equally distributed around the somata recorded (center of 2D maps), and out-of-column input would deviate in some fashion. The population 2D maps of the L6→L3 pathway support the single-cell observations: in the left ACx input was out-of-column and high-frequency-biased throughout most of the ACx and became columnar anteriorly (Fig. 3d, $n = 38$ cells, 27 mice). A comparison of the synaptic input arising from anterior and posterior sites shows that out-of-column inputs from higher frequency sites were significantly stronger than lower frequency sites (Fig. 3d, bottom columnar plot, $p = 1.4e\text{-}3$, $n = 38$, Wilcoxon rank-sum). We also found modestly significant high-frequency-biases in the inputs from L3 and L5, but not L2 and L4 (See Supplementary Fig. 2).

In the right ACx, the L6→L3 projection changed its spatial bias along the tonotopic axis: high-frequency biased posteriorly, columnar in middle sites, and low-frequency biased anteriorly (Fig. 3e, $n = 40$, 28 mice). The spatial bias of the L6→L3 input changed with the tonotopic position of the cells mapped, and on average it was not associated with a significant difference in the strength of the anterior/posterior inputs (Fig. 3e, bottom columnar plot, $p = 0.37$, $n = 40$, Wilcoxon rank-sum). The remaining laminar pathways to L3 in the right ACx were columnarly organized (See Supplementary Fig. 2).

We also quantified spatial changes of input hotspots along the tonotopic axis on a cell-by-cell basis. For this analysis, we focused on the L6→L3 pathway because this was the only projection where both auditory cortices exhibited out-of-column projections. We measured the horizontal distance between the soma and the weighted centroid of its L6 hotspot, and made scatter plots pairing the centroid with the tonotopic location of the corresponding cell (Fig. 4a, insert). The L6→L3 pathway had the steepest rate of change in tonotopic space compared to other laminar pathways examined: every 150 μm change in a cell's tonotopic position lead to an equal change in hotspot position in the left ACx, and to a 100 μm change in the right ACx (Fig. 4a, b). The absolute tonotopic position of a cell and the relative tonotopic position of its L6 input were also significantly correlated in the left ACx (Spearman $r = -0.72$, $p = 7e\text{-}7$, $n = 38$) and right ACx (Spearman $r = -0.71$, $p = 3e\text{-}7$, $n = 40$; Fig. 4; all pathways summarized in Supplementary Table 1).

To investigate in more detail, the intra- and interhemispheric tonotopic trends in the L6→L3 pathway, we examined the relationship between cortical space and tonotopy. Using a DiI-coated 16-channel silicone probe, we recorded tone-evoked multiunit activity across all layers in multiple sites across the tonotopic axis of the ACx ($n = 4$ mice, 13 recording sites; see Methods). We found that approximately one-octave change in frequency preference occurs over the span of 0.275 mm of cortical space, and approximately a three-octave change occurs in 0.5 mm (Supplementary Fig. 3). Based on these estimates, we divided our population of cells mapped in vitro into three 0.3 mm clusters named: Anterior, Middle, and Posterior. Using the soma to L6→L3 hotspot centroid measurements, we tested for significant interactions between clusters within and between the

hemispheres. In the left ACx, there was a significant intrahemispheric interaction between hotspots and clusters in tonotopic space (Fig. 4c, dashed lines; $F(2, 36) = 18.4$, $p < 7e\text{-}6$, $n = 38$, one-way analysis of variance; ANOVA). A multiple comparison analysis revealed a significant difference in hotspot location in anterior–middle (95% confidence interval (95% CI), 61.76–187.87) and anterior–posterior (95% CI, 83.61–219.40) clusters, but not middle–posterior (95% CI, $-39.92$–93.30; corrected for multiple comparisons, Bonferroni post hoc; statistics summarized in Supplementary Table 2). In the right ACx, there was also a significant interaction between hotspots and clusters in tonotopic space (Fig. 4c, solid lines; $F(2, 38) = 29.82$, $p < 6e\text{-}7$, $n = 40$, one-way ANOVA). Multiple comparisons showed significant differences in L6→L3 hotspot location for all three clusters (anterior–middle (95% CI, 46.38–326.22), anterior–posterior (95% CI, 271.2–532.96), and middle–posterior clusters (95% CI, 75.806–355.7); corrected for multiple comparisons, Bonferroni post-hoc (statistics summarized in Supplementary Table 3).

An interhemispheric analysis of variance revealed a significant interaction between L6 hotspot location and hemisphere ($F(2, 76) = 14.02$, $p < 7e\text{-}6$, two-way ANOVA, $n = 78$). A multiple comparison analysis showed that the lower frequency bias of the L6→L3 projection in anterior portions of the right ACx was significantly different from all other intra and interhemispheric clusters ($p < 5e\text{-}8$). The high-frequency bias observed in the L6→L3 projection in middle and posterior sites in the left ACx was not significantly different from posterior sites in the right ACx ($p = 0.44$). Moreover, the high-frequency L6→L3 bias in these sites was absent from anterior sites in the left ACx and middle sites in the right ACx ($p < 3e\text{-}4$). The columnar organization of the L6→L3 projection in anterior sites of the left ACx was significantly different from all other inter and intrahemispheric sites ($p < 3e\text{-}7$), except middle sites in the right ACx ($p = 0.66$; corrected for multiple comparisons, Bonferroni, post hoc). Although the latter finding suggests that the means of the L6→L3 hotspot distribution in anterior left ACx and middle right ACx are not significantly different, their spread indicates that their variance differs (Fig. 4c right). We tested this hypothesis and found that L6→L3 hotspots in middle sites of the right ACx had a significantly higher variance than anterior sites of the left ACx (F = 0.10, 95% CI (0.03–0.32), $p < 3e\text{-}4$, $n = 25$, two-sample F test). We summarized all the intrahemispheric and interhemispheric L3 connectivity motifs in Fig. 4d.

We also compared the cell density and excitability between the auditory cortices to examine whether differences in these features could contribute to lateralized activation. We found no significant difference in neuronal density between the left ACx and right ACx ($p = 0.5141$, Wilcoxon rank-sum, $n = 2$ mice). In addition, we found no significant hemispheric differences in excitability (number of action potentials per UV flash, $p = 0.24$; distance action potentials were evoked from soma, $p = 0.47$, Wilcoxon rank-sum, $n = 20$ cells for each hemisphere). At last, laminar input analysis revealed that there was no hemispheric difference in the average strength of synaptic inputs to L3 ($p = 0.31$, $n = 78$ cells, Wilcoxon rank-sum). These results are summarized in Supplementary Fig. 4.

**Axonal morphology reflects circuit asymmetries.** Pyramidal neurons in L6 are the source of the lateralized circuit-motifs we have discovered in the ACx. To examine whether the tonotopic changes in the L6→L3 pathway are reflected in the axonal arborizations of L6 pyramidal neurons we characterized their projection patterns. We performed whole-cell recordings in slices from two groups of L6 pyramidal neurons in the left ACx and

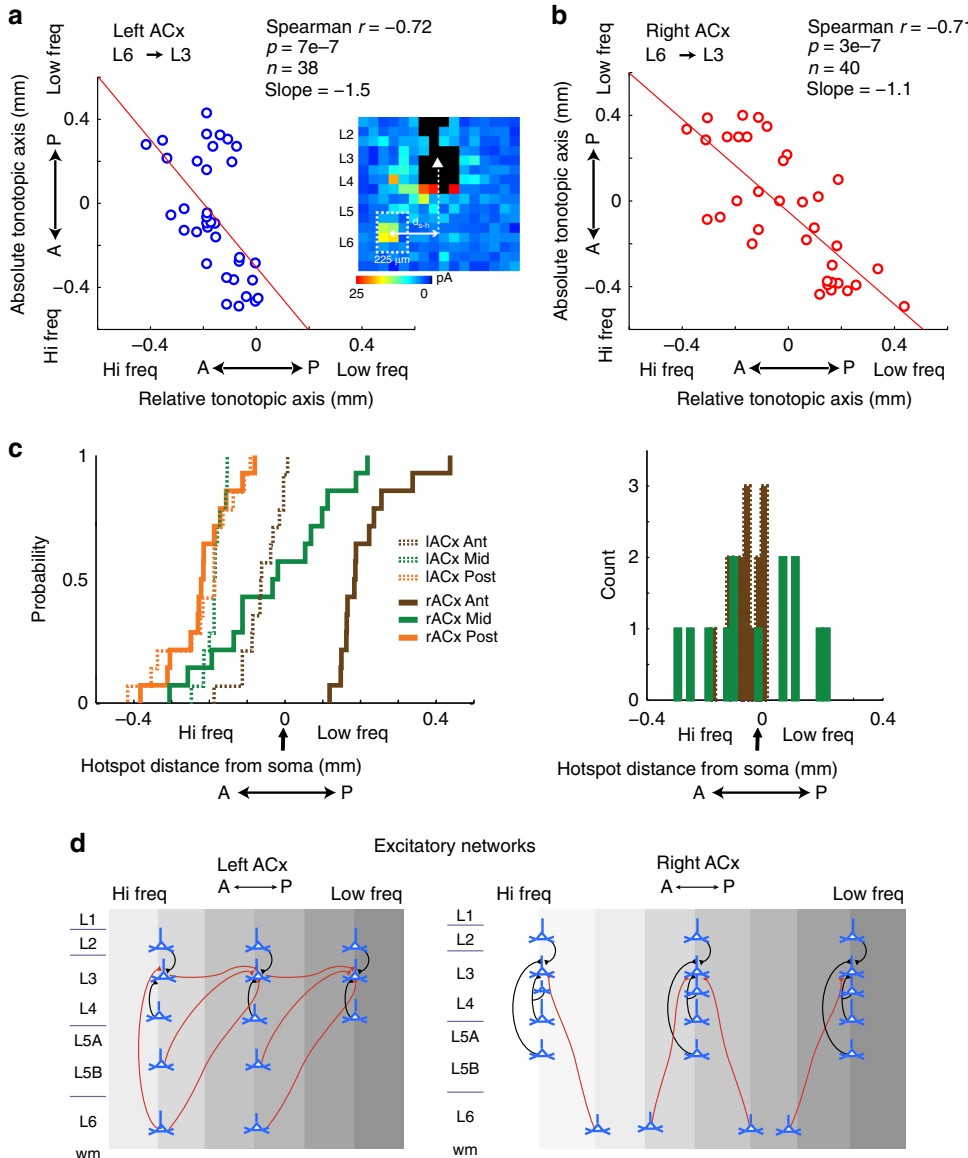

**Fig. 4** Lateralized circuit-motifs in the left and right ACx. **a, b** Population scatter plot pairing each cell's absolute tonotopic position (*y* axis) with the relative tonotopic location of its L6 hotspot centroid (*x* axis), for the left and right ACx, respectively. Red line is the best linear fit through the data. Inset depicts a single-cell average input showing how the distance from the soma to the hotspot centroid was calculated. In the *y* axis, 0 represents the tip of the hippocampus, and in the *x* axis, 0 is the position of the cell in the LSPS map grid. **c**, left Intra and interhemispheric cumulative probability density function of L6→L3 hotspot distance from soma in anterior (brown), middle (green) and posterior (orange) clusters in the left (dashed) and right (solid) ACx. **c**, right Distribution of L6→L3 hotspots from anterior sites in the left ACx (dashed brown bars) and middle sites in the right ACx (green bars). **d** Wiring diagram of excitatory networks in the left ACx and right ACx. Black lines represent columnar projections, and red lines represent out-of-column projections

filled them with biocytin for morphological reconstruction. One group of L6 pyramids was located in the most anterior portion of the ACx (corresponding to our most rostral functional mapping region), and the second group ~0.4 mm posterior to the first group. Based on our connectivity data, we predicted that the posterior group of L6 pyramids would have axonal arbors biased towards lower frequencies than their home column, whereas the rostral group would have less bias. The anterior and posterior groups of L6 pyramids reconstructed (Fig. 5a, *n* = 4 (anterior), *n* = 5 (posterior)) had qualitatively distinct patterns of axonal arborization that matched our predictions. To quantify whether these differences were significant we analyzed the two-dimensional (2D) process length densities of the anterior and posterior groups (see

Methods and Supplementary Fig. 5). We compared the fraction of axons extending anteriorly versus posteriorly relative with the soma by summing all pixels on the respective sides of the 2D population maps (Fig. 5b, c). The fraction of axons extending posteriorly was greater in the posterior group than the anterior group (*p* = 0.0124 unpaired two-tailed *t* test, *n* = 4 (anterior), *n* = 5 (posterior)). The fraction of axons and dendrites truncated in the slicing process did not significantly differ between the two groups, and was considerably lower than the fraction of intact endings in all cells (Supplementary Table 4). Although we cannot exclude the possibility that slicing artifacts masked some anatomical features, these data indicate that cells in both groups were comparable in terms of their overall preservation.

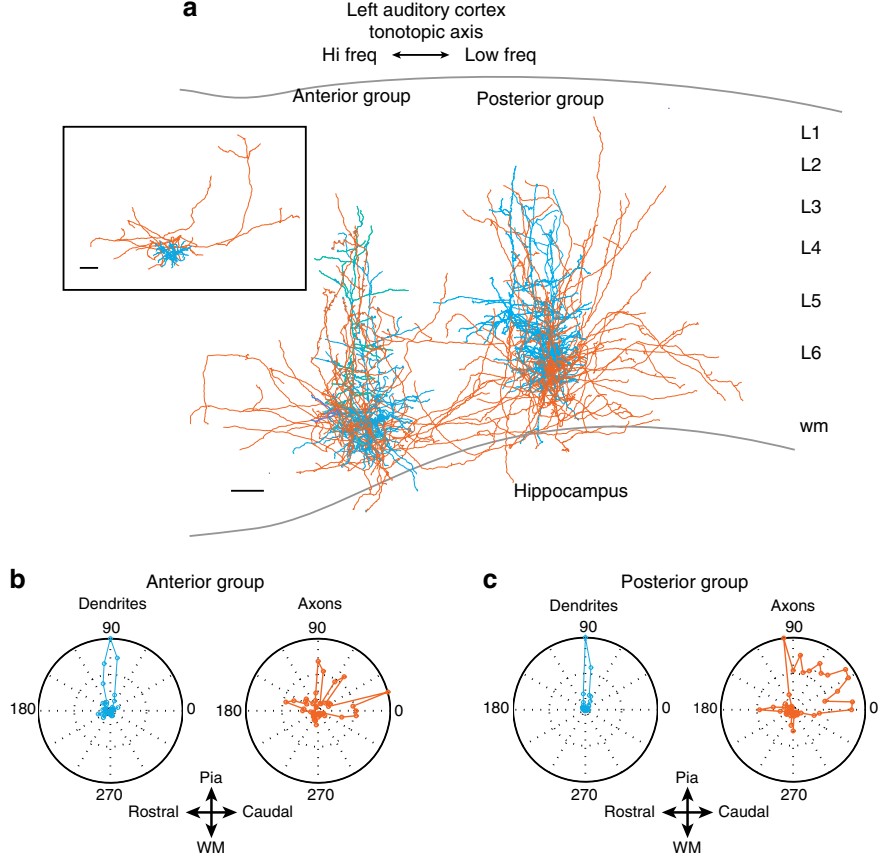

**Fig. 5** Axonal arborizations reflect circuit-motifs in the left ACx. **a** Neurolucida reconstructions of L6 pyramidal neurons in anterior ($n = 4$) and posterior ($n = 5$) regions of the ACx. Dendrites and somata are shown in blue, axons in red. Inset shows one of the cells reconstructed with out-of-column and superficial projecting axons. Scale bars 0.1 mm. **b, c** Polar plots of the processes along the tonotopic axis. 0 degrees = posterior direction, 90 degrees = pia, 180 degrees = anterior, 270 degrees = white matter

**Layer 6 subpopulation underlying asymmetric projections**. Layer 6 consists of excitatory neuronal subclasses that can project to the thalamus (cortico-thalamic, CT), the claustrum, and other cortical targets (cortico-cortical, CC). These projection patterns are in turn associated with morphological subtypes[17,18]. We tested whether CT neurons could underlie the out-of-column projections from L6 using the Ntsr1-cre driver line, which specifically targets CT neurons in L6[19,20]. We examined the identity of this pathway in the left ACx where the asymmetric projection is prevalent throughout the tonotopic axis (Fig. 3d). In the first experiment, we injected AAV-EF1a-DIO-ChR2-EYFP in Ntsr1-cre male mice and performed Channelrhodopsin-Assisted-Circuit-Mapping (CRACM[21]). We performed whole-cell, voltage-clamp recordings from L3 pyramidal neurons while minimally activating channelrhodopsin-positive cells in L6 with flashes of UV light (Fig. 6a). Half of the L3 cells mapped received L6-Ntsr1 input (10/20), and interestingly the input was organized in a columnar manner (Fig. 6b). Although we do observe columnar input from L6 with glutamate uncaging, it is significantly weaker than the out-of-column pathway (Supplementary Fig. 6). This experiment suggests that L6-Ntsr1 neurons do not underlie the out-of-column pathway.

In a second experiment to test whether the L6-Ntsr1 neurons underlie our circuit-motifs, we used chemogenetics by injecting the synaptic silencer AAV2-CAG-DIO-mCherry-2a-hM4D-nrxn1a[22] in Ntsr1-cre mice. The axon-targeted inhibitory designer receptor hM4D-neurexin was expressed in L6-Ntsr1 neurons to test whether selectively silencing this population would abolish or reduce the out-of-column L6→L3 pathway. To

assess the effectiveness and specificity of this presynaptic silencing, we first performed voltage-clamp recordings from AAV infected Ntsr1-L6 cells and used LSPS to measure synaptic input before and after bath application of clozapine-N-oxide (CNO). As expected, synaptic input arising from cells outside of L6 was not affected by application of CNO, whereas local input within L6 decreased significantly (Supplementary Fig. 7). Given that L6-Ntsr1 input can be specifically silenced with this tool, we proceeded to perform voltage-clamp recordings from L3 and map synaptic input before and after bath application of CNO (Fig. 7a). We found that the out-of-column input was not abolished or significantly impacted with the application of CNO (Fig. 7b, $p = 0.895$, $n = 5$ cells, five mice, Wilcoxon rank-sum). These results support the CRACM findings that Ntsr1 cells in L6 may not be responsible for the out-of-column circuit-motifs we describe in this study.

**Asymmetric connectivity is hearing experience dependent**. To establish a closer link between lateralized processing and asymmetric circuit-motifs, we examined the impact of development and experience on their ontogeny. First, we tested whether there was lateralized activation at hearing onset in the mouse (P12–14), by comparing Fos activity in the left and right ACx in response to vocalizations. Unlike the adult (Fig. 1), we found that Fos activity is not significantly different between the left and right ACx at hearing onset (Fig. 8a, b, $p = 0.1029$, $n = 3$ mice, Wilcoxon rank-sum). Interestingly, there was more Fos activity in L6 than L3 at hearing onset (Fig. 8c, $p = 1.1351e-04$ for the left ACx, $p = 0.0043$

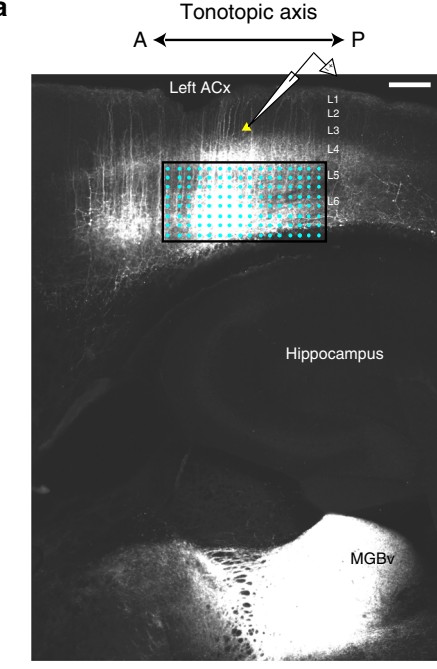

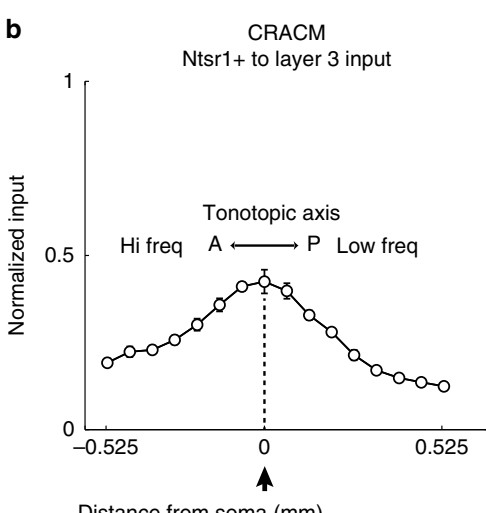

**Fig. 6** Channelrhodopsin-assisted mapping of L6-Ntsr1 input to L3. **a** Confocal image of channelrhodopsin expression in the auditory cortex. Axons are clearly seen in the auditory thalamus (MGBv). Neurons in L3 centered on the injection bolus were targeted for whole-cell recordings. Scale bar 0.3 mm. **b** Using minimal laser stimulation (at threshold to evoke a measureable postsynaptic response), we obtained the spatial profile of L6-Ntsr1 input to L3. Normalized columnar average of L6-Ntsr1 input to cells in L3 (*n* = 10) revealed input largely centered on the somata of the cells recorded. All data are presented as mean ± SEM

for the right ACx, *n* = 3 mice, Wilcoxon rank-sum), whereas in the adult upper layers had higher Fos activity (Fig. 1c, d).

Given that at hearing onset there was no lateralized activation of the auditory cortices, we wondered whether the L6→L3 out-of-column circuit motif was present at hearing onset. To answer this question, we mapped input to L3 between P12 and 14 in the left ACx using LSPS and found that the L6→L3 pathway is absent at hearing onset (Fig. 9a, *n* = 20 cells, eight mice). Input to L3 at this age largely arises from local connections and from L4. Therefore, the development of the out-of-column circuit motif appears to be

dependent on hearing experience in mice. To confirm the role of hearing experience on the development of this pathway, we raised mouse pups in an environment with white noise in the background from postnatal day 7–25. Using LSPS we mapped input to L3 in noise-reared (*n* = 25 cells, six mice) and age-matched controls (*n* = 25 cells, six mice) and found that the development of the L6→L3 circuit motif is completely disrupted when the mice are raised in white noise (Fig. 9b). In noise-reared mice, recurrent connectivity strengthens and out-of-column input does not develop. These experiments provide evidence for the functional significance of the out-of-column L6→L3 circuit motif: it is developmentally regulated (changes over time) and it is experience dependent. Columnar and local inputs are clearly present by hearing onset; therefore, the appearance of the L6→L3 out-of-column input after normal hearing experience suggests that it plays an important role in specialized auditory computations.

## Discussion
In the present study, we provide for the first time a direct link between lateralized activation, auditory function, circuits, development, and morphology. Using 3D volume imaging of Fos expression, we found a significant difference in activation between the left and right ACx (but not in the auditory thalamus) in response to mouse vocalizations and sweeps. Playing sweeps led to widespread activation of the right ACx, whereas vocalizations preferentially activated anterior portions of the left ACx where ultrasonic frequencies are represented (Fig. 1g), and both sets of stimuli produced greater activation in superficial cortical layers (Fig. 1c, d). In contrast, laminar activation was greater in deeper layers at hearing onset (Fig. 8c). This change in laminar activation fits with the fact that deeper cortical layers mature earlier than superficial layers owing to the inside–out development of neocortical layers[23].

In a previous study, we compared the synaptic organization between the orthogonal tonotopic and isofrequency axes of the left ACx[13]. We found that this functional anisotropy is reflected in the organization of cortical circuits: in slices where tonotopy was preserved L3 pyramidal neurons preferentially received out-of-column, high-frequency-biased input, whereas in slices preserving the isofrequency axis (cut coronally) these asymmetries were absent. These anisotropic circuit features in L3 of the left ACx, and the present Fos data showing significant activation in superficial cortical layers, led us to investigate function and circuit differences in these layers of the ACx. Here, we show lateralized circuit-motifs that arise from the synaptic organization of input to L3 along the tonotopic axis in horizontally cut slices. In the left ACx, several intracortical pathways emerged with modestly significant out-of-column projections: L3→L3, L5→L3 (Supplementary Fig. 2), whereas the L6→L3 pathway in the left ACx (Fig. 3d) and right ACx (Fig. 3e) had the most significant asymmetries. The properties of these out-of-column circuit-motifs differed substantially between the hemispheres. In the left ACx, input to L3 originated from higher frequency bands throughout much of the tonotopic axis (Fig. 4d, left). These consistently high-frequency biased pathways suggest that the left ACx is a specialist that facilitates the detection of specific sequences that are perhaps prevalent in mouse vocalizations. On the other hand, the L3 circuit-motifs in the right ACx had significantly different properties compared to the left ACx (Fig. 4d, right). The tonotopic location of the L6→L3 input systematically shifted its spatial bias throughout the tonotopic axis. Even when the L6→L3 pathway transitions to a columnar organization in middle portions of the right ACx, its location was significantly more variable compared with the distribution of columnar

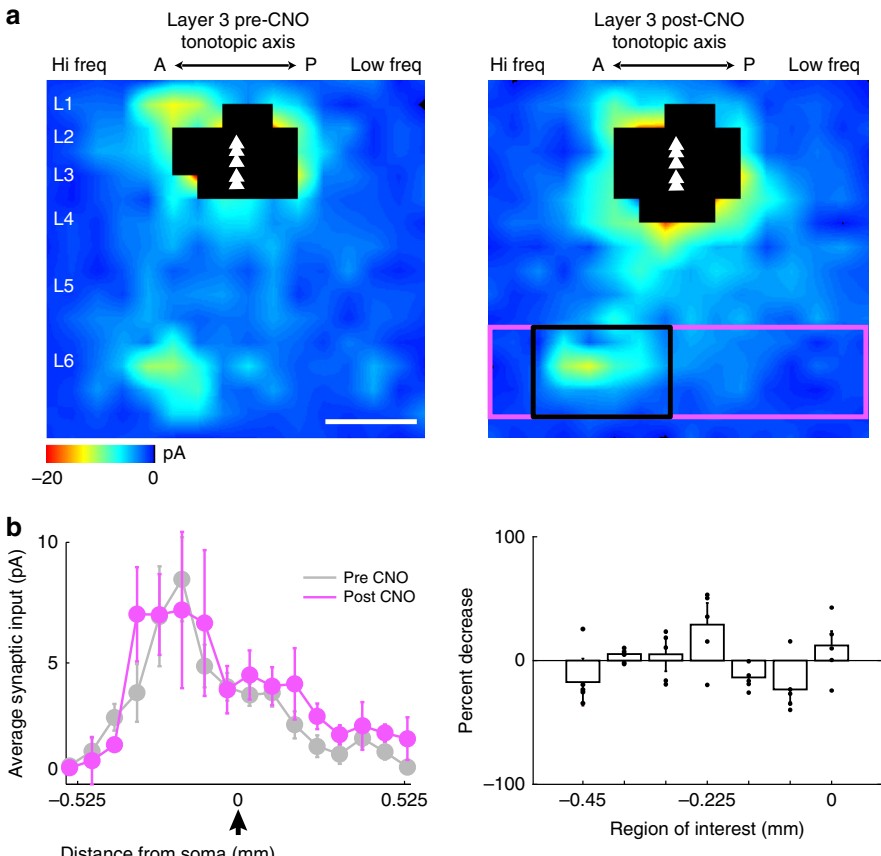

**Fig. 7** Chemogenetic silencing of L6-Ntsr1 input to L3. **a** Average population input maps of L3 neurons before (left) and after (right) presynaptic silencing of Ntsr1 input with CNO infusion into the bath. Scale bar 0.3 mm. **b**, left Columnar average of synaptic from L6 (boxed pink region in panel A right) to L3 before and after CNO infusion (*n* = 5 cells, five mice). **b**, right The percent decrease in out-of-column L6 input (boxed black region in (**a**) right) to L3 was largely unchanged after CNO infusion. All data are presented as mean ± SEM

L6→L3 input in anterior portions of the left ACx (Fig. 4c, right). A study of the rat's right ACx showed that there is a map of sweep direction selectivity in parallel with the tonotopic axis[12]. In that study the skewness (a measure of asymmetry) of tone-evoked excitatory synaptic potentials changed monotonically with BF: neurons with low best frequencies had a long tail of weaker responses at higher frequencies, neurons with high best frequencies had a long tail of weaker responses at lower frequencies, and neurons with intermediate frequency tuning were symmetrical. The tonotopic shifts in the L6→L3 pathway we discovered in the right ACx could underlie these patterns of asymmetric excitatory potentials. Furthermore, we found that the L6→L3 projection in middle sites of the right ACx (in putative intermediate frequency bands) was not strictly columnar, but instead appeared to arise from variable frequency bands around the home column, potentially facilitating mixed sweep selectivity (Fig. 4c, right).

In both hemispheres, the out-of-column projections were horizontally shifted up to 400 μm away from the home column of the cells mapped (Fig. 4c). The spatial scale of these pathways suggests integration over at least 2-3 octaves in the mouse ACx (Supplementary Fig. 3). Moreover, L3 neurons in both hemispheres appear to be integrating multiple sources of spectral input: home-column frequency input (via direct thalamic input or local L4 input) and multiple sources of spectrally biased input (Fig. 4d). This combination of columnar and out-of-column spectral input bolsters the idea that superficial layers in the ACx are involved in the processing of spectrally complex sounds,

which may require integration across frequency bands by the underlying circuits[24–27]. A growing number of studies in vivo also support this proposal that integration of multiple pathways in superficial layers might underlie the observed sensitivity to several sound features[28,29]. The bandwidth of frequency receptive fields in the ACx are much broader than subcortical auditory areas, can span much of the audible spectrum, and is not the result of broad thalamic input[30]. Evidence for asymmetric facilitation has been reported in responses to frequency modulated sweeps in the ACx of the pallid bat, where it was reported that a population of auditory cortical neurons was sensitive exclusively to downward sweeps[27]. One way to accomplish this sweep selectivity is to detect appropriately timed tone sequences that will generate firing. A proposed connectivity scheme involves a population of output neurons receiving input from cells in different frequency channels (e.g., a combination of home-column input and out-of-column frequency input[26]). The direction and magnitude of the tonotopic bias from the out-of-column input would determine the sweep rate and direction selectivity of the output neurons. The specialist and generalist L6→L3 circuit-motifs uncovered in the ACx of our present (Fig. 4d) and previous study[13] support this model, and could potentially underlie the connectivity schemes to detect specific tone sequences. Lateralized circuit-motifs are not only found in excitatory neuronal populations, but also in different inhibitory cell types. In the left ACx we found that parvalbumin (PV) and somatostatin (SOM)-positive inter-neurons largely received local recurrent connections, in agreement with what has been reported in other sensory cortical

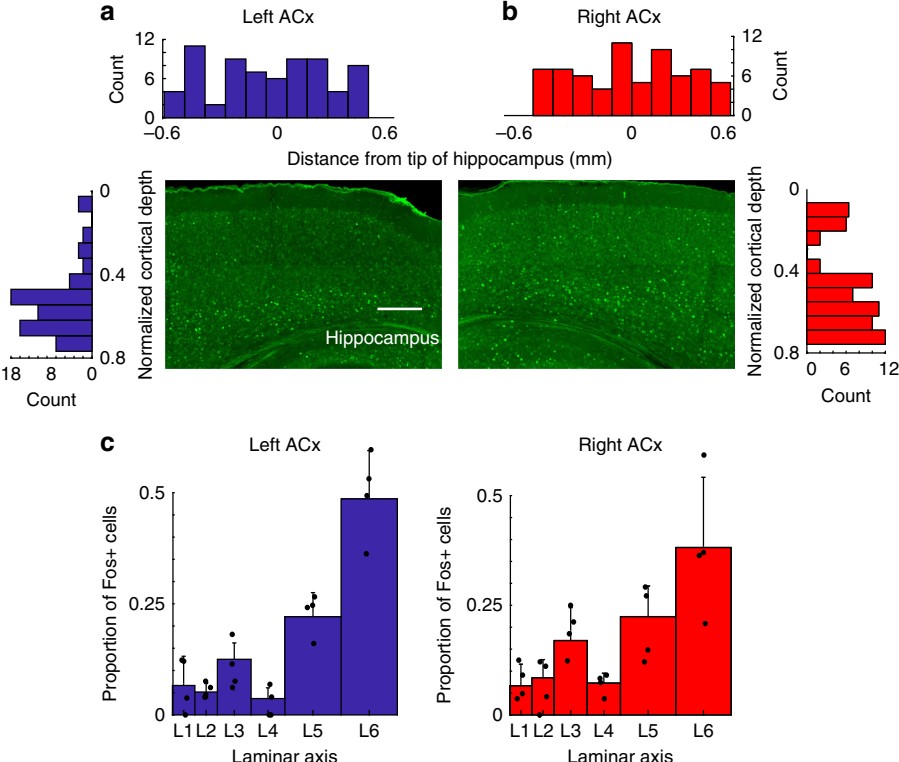

**Fig. 8** Hearing onset Fos activity in response to vocalizations. **a, b** Quantification of immuno Fos staining data at hearing onset (P12–14) showing that there is no difference in the number of Fos-positive neurons between the left and right ACx. Scale bar 0.2 mm. **c** Population plots showing that although there is no difference in the total number of active cells between the hemispheres, there is a significant difference in the activation of superficial and deep layers ($n = 3$ mice). All data are presented as mean ± SEM

areas[31]. In contrast, in the right ACx PV+ and a subpopulation of SOM+ interneurons received strong infragranular input[32]. These significant connectivity differences could have a profound effect on how cortical gain control is achieved[33].

The CRACM and chemogenetic experiments (Fig. 6–7) suggest that Ntsr1-L6 neurons do not significantly contribute to the out-of-column pathway. Notwithstanding, L6 has a diverse population of principal neurons that could underlie this projection. Potential candidates are CC neurons, which have long horizontal axons, diverse dendritic morphology, phasic firing, and project to higher-order auditory areas[17,18,34]. The morphology of some of the cells we reconstructed in L6 with out-of-column projecting axons is consistent with CC neurons (Fig. 5a, inset). In future experiments, we intend to identify genetic markers for this out-of-column projecting population of L6 neurons. At hearing onset the out-of-column pathway in the left ACx is absent (Fig. 9a), and there is no lateralized activation in response to vocalizations (Fig. 8). These findings suggest that the emergence of auditory processing features is influenced by auditory experience. To reinforce this link we examined the impact of noise rearing on the L6→L3 pathway. The effect of noise on delaying tonotopic map refinement in the ACx is well known[35], but not how it directly impacts synaptic connectivity. We found that it prevents the stabilization of experience-dependent circuit-motifs, and unexpectedly strengthens input from L4 (Fig. 9b). These changes suggest that the noise-reared circuit is not completely analogous to the immature hearing onset connectivity, but instead develops its own idiosyncratic circuitry. In future studies, we plan to use sequencing techniques to genetically identify and target out-of-column projecting neurons to test directly their role on lateralized functions.

Our results strongly support the idea that the acoustic properties of species-specific vocalizations shape neural circuits in an experience-dependent manner. More specifically, mouse vocalizations contain syllables with fast and stereotyped frequency jumps. In particular, downward pitch jumps are the most prevalent components in their syllables[36]. Our in vivo sweep direction selectivity data show that the left ACx has a widespread preference for downward sweeps, and circuit-motifs that could facilitate their detection. Perhaps the prevalence of downward pitch jumps in mouse vocalizations, combined with experience-dependent malleability of cortical circuits underlie the functional and connectivity specializations we report in the left ACx. More broadly, the lateralized circuit-motifs and spectrotemporal selectivity we report could support the Asymmetric Sampling Theory of speech. This theory proposes that the left ACx performs analyses requiring high temporal resolution (e.g., formant transitions in speech), and the right ACx performs analyses requiring high spectral resolution (e.g., speech intonation)[37].

## Methods

**Experimental mice.** Experiments used CBA/J mice (Jackson Labs) in strict accordance with the National Institutes of Health guidelines, as approved by The City College of New York Institutional Animal Care and Use Committee.

**Sound delivery and iDISCO with Fos immunohistochemistry.** Individual CBA male mice aged postnatal days (P) 45–60 were placed in a double-walled sound booth in their home cage (with food, water, and air circulation) to acclimate for 5 hours to achieve baseline levels of Fos expression in the ACx prior to sound presentation. Mice were presented with free-field stimulation (Avisoft Bioacoustic speakers, Glienicke, Germany) of either mouse vocalizations (downloaded with permission from the Pasteur Institute's mouseTube) or frequency modulated sweeps (up and down sweeps of 1–4 kHz at 2 octaves sec$^{-1}$). Both stimulus sets

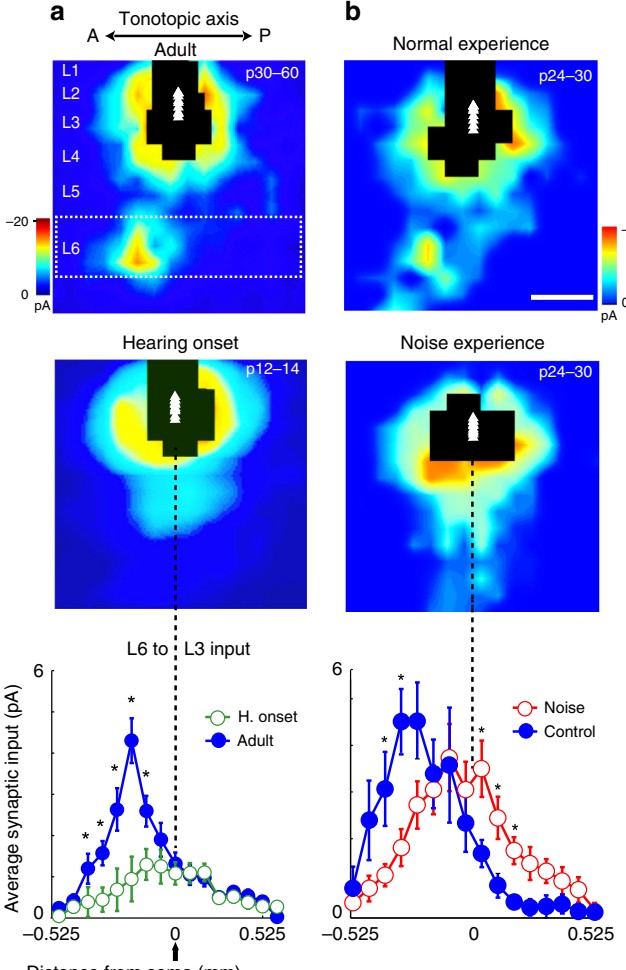

**Fig. 9** Development and experience-dependence of the L6→L3 out-of-column circuit motif. **a** Comparison of synaptic input to L3 in the ACx of adult (top, P30–60; $n = 12$ cells, 10 mice) and hearing onset mice (middle, P12–14; $n = 20$ cells, eight mice). The lack of out-of-column input from L6 (boxed region) at hearing onset is significant (bottom, data points marked with stars are significantly different). **b** Comparison of synaptic input to L3 in age-matched control mice (top, P24–30, $n = 25$ cells, six mice) and noise-reared mice (middle, P24–30, $n = 25$ cells, six mice). Noise rearing significantly impacted the formation of out-of-column input from L6 (bottom, data points marked with stars are significantly different). Scale bar 0.3 mm. All data are presented as mean ± SEM

were 2 s long with an 18 s inter-stimulus interval and were presented for 30 minutes. To allow time for the expression of Fos, mice were perfused 50 minutes post-sound stimulation. Mice were perfused with 4% paraformaldehyde and their brains were processed with the iDISCO protocol with Fos immunostaining (sc-52, 1:200, Santa Cruz) with the secondary antibody donkey anti-rabbit Alexa 647 (1:500, Life Technologies).

**Light sheet imaging.** Cleared samples were imaged in the sagittal orientation (right lateral side up) on a light sheet microscope (Ultramicroscope II, LaVision Biotec) equipped with a sCMOS camera (Andor Neo) and a × 2/0.5 objective lens (MVPLAPO × 2) equipped with a 6 mm working distance dipping cap. Version v144 of the Imspector Microscope controller software was used. The microscope is equipped with LED lasers (488 nm, 561 nm, and 640 nm) with three fixed light sheet generating lenses. Scans were made at the × 0.8 zoom magnification (× 1.6 effective magnification), with a light sheet numerical aperture of 0.1. Emission filters used were 525/50 and 680/30. The samples were scanned with a step-size of 3 μm using the continuous light sheet scanning method with the included contrast blending algorithm for the 640 nm channel (20 acquisitions per plane with a 50 ms exposure), and without horizontal scanning for the 480 nm channel (50 ms

exposure). To speed up the acquisitions, both channels were acquired in two separate scans. To account for micro-movements of the samples that may occur between the scans, a 3D image affine registration was performed to align both channels using ClearMap.

**ClearMap analysis.** All analysis and quantifications were performed with our open source ClearMap software as shown in the example scripts, and the associated documentation and as described in reference[14]. The latest version of ClearMap, scripts and documentation can be obtained from http://www.idisco.info. The settings used for analyzing the data were as follows: the background was removed by subtraction of the morphological opened image with a disk shape structure element with main axis of seven pixels in diameter. Cells were detected from peaks and subsequent watershedding, removing background pixels below an intensity cutoff of 700 (700 is the rounded average value of the background intensity after the background subtraction) and selecting cells with sizes between 20 and 500 voxels. Density maps were generated by summing spheres of 375 μm diameter (=15 pixels) and uniform intensity centered on each cell. Samples were registered using the average autofluorescence STPR brain registered to the Allen Brain Institute 25 μm map, and its companion annotation map (http://alleninstitute.org/). Non-parametric $t$ tests were used to calculate the $p$ values (significance level set to 0.05) of the difference of the means of the heatmap voxels between each group.

**In vivo recording methods: surgery.** We made a craniotomy (2 mm by 3 mm, centered at 2.5 mm posterior, and 4 mm lateral to Bregma) on the left and right auditory cortices of anesthetized (75 mg kg$^{-1}$ ketamine, 0.5 mg kg$^{-1}$ medetomidine) CBA/J male mice aged P28–40 ($n = 23$ mice) to perform cell-attached recordings (10–50 MOhm seal). We targeted neurons in the same area and layers characterized in vitro (L2/3 140–300 μm below the cortical surface) using patch electrodes containing the same intracellular solution described in the section below.

**Stimuli.** A sound booth (Industrial Acoustics Company, Bronx, NY) was used to conduct all recordings. We used a custom built real-time Linux system (200 kHz sampling rate) driving a high-end Lynx L22 audio card (Lynx Studio Technology Inc., Newport Beach, CA) with an ED1 electrostatic speaker (Tucker-Davis Technologies, Alachua, FL) in free-field configuration (speaker located 8 cm lateral to, and facing, the contralateral ear). The stimuli used were 100 ms long pure tones of 16 frequencies logarithmically spaced between 1 and 40 kHz, presented at 5, 20, 50, and 80 dB SPL, and logarithmic frequency modulated sweeps between 1 and 40 kHz with speeds of 25–150 octaves s$^{-1}$, at 60 dB SPL. All tones and sweeps were presented in a fixed pseudorandom sequence at a rate of one tone or sweep per second.

**In vivo recording analysis.** To quantify changes in the firing rate during pure tones, we divided the responses into 10 25-ms epochs, which included 50 ms prior to the stimulus and 100 ms post. We only analyzed tone-responsive neurons: epochs with a mean firing rate three standard deviations above background (pre-stimulus). The best frequencies reported in the manuscript were calculated by adding the spike count across all sound levels and finding the frequency that evoked the highest firing rate. To quantify sweep direction selectivity, we selected the sweep speed that evoked the highest peak firing rate using a sliding window algorithm (5 ms bins). Direction selectivity was defined as $(R_{up\_sweep} - R_{down\_sweep})/(R_{up\_sweep} + R_{down\_sweep})$, where $R$ is the peak firing rate evoked at the preferred sweep speed.

**Silicone probe recordings.** We made a craniotomy (2 mm × 3 mm, centered at 2.5 mm posterior and 4 mm lateral to Bregma) on the left and right ACx of anesthetized (30 mg/kg ketamine, 0.24 mg/kg medetomidine) CBA/J mice aged P40 ($n = 4$ mice, 13 total recording sites) to perform translaminar multiunit recordings. A Neuronexus (Ann Arbor, MI) silicone probe (A1×16–50) was used to record multiunit activity in the ACx. The probe was aligned parallel to the tonotopic axis. The stimuli used were 100 ms long pure tones of 20 frequencies logarithmically spaced from 1 to 46 kHz, presented at 10, 30, 50, and 70 dB SPL. All tones were presented in a fixed pseudorandom sequence at a rate of one tone per 500 milliseconds. To further corroborate the location of our in vivo recordings and to confirm that the anatomical location of our horizontal slice recordings were from primary ACx, we marked our recording sites with DiI (Invitrogen).

**Histology.** We perfused the mice using 4% paraformaldehyde in phosphate buffer and sliced the brain in a horizontal cut to recover the DiI tracks of our recording sites along the tonotopic axis.

**Analysis of silicone probe recordings.** To quantify changes in the firing rate during pure tones, we divided the responses into 10 25-ms epochs, which included 50 ms prior to the stimulus and 100 ms post. We report the instantaneous firing rate (25 ms window) of tone-responsive epochs with a mean firing rate three standard deviations above background (pre-stimulus). In every animal we measured BF preference using a silicone probe in two to six locations within the ACx. We then marked with a DiI track locations that were one octave and three octaves

apart. The mice were perfused and the tracks recovered. The octave-to-mm factor we report in the main text is the average across mice of the cortical distances between DiI tracks recovered.

**Slice preparation and electrophysiology**. We used CBA/J male mice aged P30–40. Mice were anesthetized and decapitated and the brains were transferred to a chilled cutting solution composed of (in mM): 110 choline chloride, 25 NaHCO$_3$, 25 D-glucose, 11.6 sodium ascorbate, 7 MgCl$_2$, 3.1 sodium pyruvate, 2.5 KCl, 1.25 NaH$_2$PO$_4$, and 0.5 CaCl$_2$. We made horizontal slices to examine synaptic connectivity along the tonotopic axis. We sliced using a 15-degree angle between the blade and the medial-lateral axis to obtain apical dendrites parallel to the slice in the ACx[38]. All slices were 300 μm thick and were transferred to artificial cerebrospinal fluid (ACSF) containing (in mM): 127 NaCl, 25 NaHCO$_3$, 25 D-glucose, 2.5 KCl, 1 MgCl$_2$, 2 CaCl$_2$, and 1.25 NaH$_2$PO$_4$, aerated with 95% O$_2$ 5% CO$_2$. The slices were incubated at 34° for 20–30 minutes and then kept at room temperature during the experiments. Excitatory neurons 50–80 μm below the surface of the slice were visualized using infrared gradient contrast optics and patched with electrodes (6–7 MOhm) containing the following intracellular solution (in mM): 128 K-methylsulfate, 4 MgCl, 10 HEPES, 1 EGTA, 4 NaATP, 0.4 NaGTP, 10 Na-phosphocreatine, and in some cases 0.015 Alexa-594 (Molecular Probes, Eugene, Oregon, USA); (pH 7.25); 300 mOsm. The Alexa-594 in the internal solution allowed us to visualize cells to confirm that their dendritic shaft was parallel to the slice and that they were excitatory by visualizing their dendritic arbor and spines. Whole-cell recordings were made using a Multiclamp 700 A amplifier (Axon Instruments, Molecular Devices, Sunnyvale, California, USA). To map synaptic input, we measured excitatory currents at a holding potential of –70 mV, and to map excitation profiles (see below) action potentials were recorded in cell-attached configuration. We used the custom software package ephus[39] (http://www.ephus.org) for instrument control and acquisition written in Matlab (MathWorks, Natick, MA, USA).

**LSPS by glutamate uncaging**. Brain slices were prepared and maintained[40,41] in vitro as described in the preceding section, except that the ACSF was supplemented with (in mM): 0.2 nitroindolinyl (NI)-caged glutamate (Tocris), 0.005 CPP (Tocris), and a final concentration of 4 CaCl$_2$ and 4 MgCl$_2$. To accomplish focal photolysis (UV flash) of the caged glutamate compound, we used a 1 ms light stimulus consisting of 100 pulses from a pulsed UV laser (wavelength, 355 nm with a repetition rate of 100 kHz; DPSS Lasers, Santa Clara, California USA). A 16 × 16 stimulus grid with 75 μm spacing was used for LSPS mapping of synaptic input in mouse cortical slices. This configuration resulted in a mapping region of 1.125 × 1.125 mm. To avoid revisiting the vicinity of sites recently stimulated, the pattern of stimulation was pseudorandom. Each trial included a test pulse to measure electrophysiological parameters, and UV flashes were presented every 1 s. Laser power between 30 and 35 mW evoked reliable synaptic responses in all neurons recorded.

To consistently align the stimulus grid for each cell recorded, the x axis of the grid was centered on the soma and the y axis was aligned with the second row of the grid placed on the L1/2 border. We chose the latter because it is the most prominent laminar boundary in the ACx. We measured the following spatial coordinates for each cell: distance from the soma to the pial surface, L1/2 border, and the horizontal distance to the anterior tip of the hippocampus (where the fimbria exits). The latter measurement was our center reference point (x = 0) to assign each cell to either an anterior, middle or posterior cluster within the ACx. The size of the ACx in the mouse is ~ 1.5 mm[42], therefore we mapped cells located at 480 ± 42 μm (mean ± standard error of the mean (SEM)) anterior and posterior to our center reference (x = 0, Fig. 3a) in both hemispheres.

To measure the excitability across layers (i.e., number of APs per UV flash) and to measure how far from the soma a UV flash can evoke an action potential (AP), we performed cell-attached recordings to detect APs. We sampled cells from all layers to measure excitation profiles (n = 20 for each hemisphere). To construct excitation profiles, we used an 8 × 8 grid with 50 μm spacing for L2, L3, L4, and L6 neurons and for L5 pyramidal neurons we used an 8 × 16 grid with the same spacing to test for dendritically evoked spiking. APs within 50 ms of the UV flash onset were included in the analysis. The same laser power was used for synaptic maps and excitation profiles (30-35 mW).

**Analysis of LSPS data**. The mean current amplitude of synaptic input responses was calculated in the 50 ms epoch after the direct response time window (7.5 ms after UV stimulus). Note that the lack of input around somata in superficial layers comes from the exclusion of direct responses in our analyses. The values for each site stimulated are represented as pixels in a colormap. We recorded two to four maps for each cell to create an average input map, and these average maps were used for group averages and for all analyses.

**Two-dimensional (2D) tonotopic analysis**. Average input maps from each cell were subdivided into layers (see description below) and the input from individual layers was summed across rows to obtain a presynaptic input vector. Vectors from each cell were grouped in 50 μm bins, averaged, and organized by tonotopic position to produce the final 2D maps. The y axis represents the absolute tonotopic position of the cells mapped and the x axis the relative (within area mapped) tonotopic position of presynaptic input.

**2D tonotopic input significance analysis**. The 2D population maps from each layer were divided in half with columns 1–8 corresponding to input arising anterior to the soma and columns 9–16 to input arising posterior to the soma. We obtained the peak input values along each row for the anterior and posterior halves of the 2D maps and used these values to perform Wilcoxon rank-sum statistical tests.

**Hotspot analysis**. A presynaptic input hotspot was defined as contiguous pixels two standard deviations above background. The hotspot distance from soma is the horizontal distance from the hotspot's weighted centroid to the soma. Laminar boundaries were determined for every slice. The L3/L4 boundary was determined from cytochrome oxidase staining density[43], and L2/L3, L4/5, and L5/L6 differ in cell density and morphology readily visible under the differential interference contrast optics used in our experiments. The markings in each figure represent the averages of the thicknesses measured (for details, see ref. [13]). Interpolated maps of population data are for display purposes only and were not used for analysis.

**Viral injections and noise rearing**. Anesthesia was performed as described above for in vivo recordings. For the CRACM slice experiments, we injected AAV-EF1a-DIO-hChR2-EYFP (a generous gift from Cold Spring Harbor Lab) in the left ACx of p21 Ntsr1-cre (Gensat) male mice. We used a UV laser for optogenetic stimulation. For the chemogenetic slice experiments, we injected AAV2-CAG-DIO-mCherry-2a-hM4D-nrxn1a (a generous gift from Tiago Branco, Sainsbury Wellcome Centre) in the left ACx of p21 Ntsr1-cre mice. For synaptic silencing we bath applied CNO (Enzo Life Sciences) at a concentration of 20 μM. Noise-reared pups (from P7–25) were placed with their dam in a double-walled sound booth in their home cage (with food, water, and air circulation). We maintained their normal 12-hour light/dark cycle, and presented continuous 60 dB white noise.

**Morphometric analysis**. Cells in 300 μm thick acute slices were recorded in whole-cell mode as described above except that the intracellular solution additionally contained 0.5% biocytin, and the temperature in the recording chamber was held at 32–34 ºC. Cells deeper than 50 μm from the top of the slice were patched, filled for > 25 min, fixed by immersion in 4% paraformaldehyde/ 1× phosphate-buffered saline at 4 C for 3 to 14 days, and then visualized using standard avidin-biotin and 3,3'-diaminobenzidine/horseradish peroxidase histochemistry (ABC Elite kit, Vector labs, Burlingame, CA). Slices were mounted in aqueous medium (Fluoromount G, Southern Biotech, Birmingham, AL). Biocytin-labeled cells were reconstructed under a × 100 oil-immersion objective using the Neurolucida system (MBF Bioscience, Williston, VT). The raw three-dimensional reconstruction data were then analyzed using custom-written Matlab software to generate 2D process density maps[44–47]. In brief, each reconstruction was projected onto a plane (X = tonotopic axis; Y = laminar axis) by summing over the isofrequency axis. We assigned the soma a location of [X, Y] = [0, 0], divided the surrounding cortical area into pixels of 20 × 20 μm extending bidirectionally for 1 mm from the soma on each axis to yield a 100 × 100 pixel matrix, and measured the total axonal or dendritic length (ignoring process thickness) contained in each pixel. Population maps were made by summing the individual maps. p values were based on unpaired two-tailed t tests of the pixel data. For the polar histograms (Fig. 5b, c), each reconstruction was divided into 7.5 degree wedges centered on the soma. Weighted axonal and dendrite length within each wedge was then calculated by multiplying the length of each segment by its distance from the soma, so as to assign greater significance to those processes that extend far from the soma as opposed to the extensive local arbors. The resulting values were then normalized to the largest value for each data set.

**Statistical analysis**. Analyses of significant differences were performed using Wilcoxon rank-sum, one-way, and two-way ANOVA using Bonferroni's test for post hoc analysis. Significance was defined as $p < 0.05$. Correlations were calculated using Spearman's rank-order. The statistical power of the chemogenetic data (Fig. 7, and Supplementary Fig. 6) was >80% at an alpha level of 0.05. All data points are plotted ± the SEM unless otherwise noted.

**Reporting summary**. Further information on research design is available in the Nature Research Reporting Summary linked to this article.

## Data availability

The data that support the findings of this study are stored in CCNY servers and are available from the corresponding author upon reasonable request.

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

## Acknowledgements

We thank Anthony M. Zador for support in the early stages of this project, Barry Burbach, Zach Saccomano, and Sang-geol Koh for invaluable technical help, and Jonathan Levitt for use of the Neurolucida system. All LSPS hardware control and data acquisition was performed using ephus (http://www.ephus.org). The project described was supported by Grant Number 5G12MD007603 (H.V.O.) from the National Institute On Minority Health and Health Disparities. The content is solely the responsibility of the authors and does not necessarily represent the official views of the National Institute on Minority Health and Health Disparities or the National Institutes of Health. This project was also funded by a Whitehall Foundation Research Grant (H.V.O.), and an NSF Career Award (IOS-1652774, H.V.O.).

## Author contributions

H.V.O. conceived the experiments, conducted the in vivo and part of the in vitro experiments, analyzed these data and wrote the manuscript. N.R. imaged and analyzed the Fos data. R.B.L. conducted iDISCO Fos experiments, performed all the Neurolucida reconstructions and analyses, and conducted the peri-hearing in vitro experiments. A.P.R. helped conceive the optogenetic and chemogenetic experiments and performed viral injections. C.M.P. conducted the peri-hearing Fos experiments. T.M. conducted the noise rearing experiments.

## Additional information

**Competing interests:** The authors declare no competing interests.

