## [Transparent Peer Review File · Nature Communications]

Reviewers' comments:

Reviewer #1 (Remarks to the Author):

Here Levy et al address an intriguing lateralization in trans-laminar circuitry in auditory cortex that appears to be related to functional differences in sweep selectivity. The question is interesting, and the authors bring an impressive series of experiments to bear on it. The results are mixed, with some compelling findings and some negative results. Overall it is an interesting story, but I do have some concerns.

Figure 1

The heat maps are visually appealing but make it difficult to see the laminar differences. It is certainly not clear that this data has cellular resolution. There is also no quantification, the laminar differences are just asserted. The format and analysis in Fig. 7 is much better — the histograms allow a quantitative comparison and the cellular resolution is easily seen from the image. It's unclear why Figs. 1 and 7 use completely different format and analysis, since they are essentially the same type of experiment. Since the two figures are meant to be directly compared they need to be in the same format.

Fig. 1 a,b,c: The lateralization for sweeps and vocalizations looks pretty clear but the dominance of superficial layers is less obvious.

Fig. 1 d,e,f: Why is the highest FOS activity seen in somatosensory cortex?

For the anterior activation for vocalizations, the same issue arises as the L/R/laminar result above - the results should include quantification, not just "mostly anterior" or "highest activation was largely in superficial."

Fig. 2e,f is interesting and convincing.

Fig. 4

Do 4 a,b and c,d really belong in the same figure?

In 4c there's no quantification at all, just 2 examples and a statement about the octave-to-mm factor. How many recording sites? How did you arrive at this octave-to-mm factor? This could conceivably go in the methods.

Many aspects of Figure 4 are quite confusing. 4A: What is the Relative tonotopic axis (mm)? What does 0 on this axis correspond to? What do positive and negative directions correspond to? Is it the same as "Hotspot distance from soma (mm)" in panel D, or is that something different? In the histogram inset to panel D, what is the x axis? In panels A-B, on the absolute tonotopic axis, what does 0 correspond to and what are the positive and negative directions? How do these axes relate to those on the LSPS maps in Figure 3? In panel E is positive to the right and negative to the left? It would be helpful to include A-P and Hi-Low freq on all of these plots, and to standardize the axis labeling terminology so that all panels can be directly compared (or not if they shouldn't).

In the text description of the results for panel 4D, it's not clear what a cluster is and whether that is the same thing as the 3 Anterior, Middle, Posterior groups.

What does the ANOVA analysis described in conjunction with panel 4D tell us beyond what was already shown in the linear regressions in panels A,B? Is it a distinct finding or just another way of statistically testing the same question? Explain.

Figure 5b

The polar plots are unclear and the data are not convincing of a difference in axonal projection patterns. What does the polar angle correspond to? What does 0 degrees correspond to? What is

the polar radius? The fills in 5a make it look like cartesian coordinates of the A-P tonotopic axis and the pia-WM depth axis would be appropriate for analysis, but I can't understand how that relates to the polar plots. Are these cells patched in slices? How do you know the important axons aren't cut off in slicing? A single slicing artifact outlier could greatly skew the average of just 5 cells.

Figure 6

When you introduce the synaptic silencer (hM4D-nrxn1a), you should state up front whether it acts pre- or post-synaptically.

Why is the positive control in 6c,d analyzed/plotted differently than the negative result in 6e? Either report both in terms of % decrease, or in absolute pA. I assume that the rank-sum tests were performed on the same type of data, rather than the positive control on % change and the negative result on pA. A negative result with a sample size of 5 should include some sort of power analysis to assess the likelihood of a false negative due to small n.

Figure 7.

FYI, the expression is invisible when the figure is printed out.

The axes are not labeled, I am guessing that the numbers refer to cortical depth and distance along the tonotopic axis. Why are the layout and analysis completely different from that in Figure 1? I prefer the format in Fig. 7 because it includes some quantification, whereas in Figure 1 the reader can only eyeball the Fos expression. I don't understand why the two figures are formatted completely differently, it prevents a direct comparison between the two, although the main point of Fig. 7 is that it's different from Fig. 1.

Fig. 8, p.16 - You should clarify to what extent your results can distinguish between the L6→L3 connection being maturational (develops purely as a function of age) and experience-dependent. It seems to me the noise-rearing experiment shows it is experience dependent, but under normal rearing conditions it isn't clear whether its appearance depends on the onset of sensory input or on maturational age.

Fig. 8 should include the ages on the panels.

Discussion:

The inside-out development of cortical layers should be supported with a citation.

You speculate that left ACx could be specialized for detecting specific sequences that are perhaps prevalent in mouse vocalizations. This is a fairly empty statement, but could be greatly strengthened by relating your findings to the growing literature on mouse vocalizations. Which sequences are actually prevalent in mouse vocalizations that left ACx would be specialized for?

It seems like the killer experiment would be to block the L6→L3 projection in right ACx, to see if it abolishes FM sweep selectivity there. Ntsr1 cells didn't work here, so how might you do it? I think this would be a cool future direction to include in the discussion, if you're OK disclosing that.

You should discuss how your results fit into the framework laid out by Metherate (for review see Metherate et al. 2005, Hearing Research). There are a whole series of papers there that are highly relevant.

Reviewer #2 (Remarks to the Author):

Levy et al. demonstrate differences in left and right auditory cortex of mice in respect to responses to sound, layer 6 to layer 3 connectivity, and axonal projections. Furthermore, they show that the

emergence of at least some of these differences depends on normal early acoustic experience. The study is an impressive tour de force incorporating a very large array of techniques and covering a wide variety of circuits aspects. Together the results presented in this study are a significant step towards understanding the mechanisms that give rise to lateralization of function in auditory cortex.

Specific comments:

Due to the differences in interlaminar connectivity along and across the tonotopic axis (Oviedo et al., 2010), the exact slice orientation relative to the tonotopic axis is a crucial factor that influences the interlaminar connectivity observed in the present scanning and axonal reconstruction experiments. Since slight variations inevitably occur in each preparation, data need to come from a larger number of animals to reduce this influence. This the authors should state the number of animals included for every experiment, not only the number of cells. In order to exclude the possibility that slice orientation is not a contributing factor, I also suggest to analyze whether 'columnar offset' of L6 to L3 shows some subject correlations.

Figure 3 A: The lower panel photo is identical to the photo the senior author has previously published in Nat. Neuroscience 2010. Using the same photo twice is considered self-plagiarisms. The lookup tables change for each example cell, which makes it difficult to compare the maps. I assume peak amplifies in pA are shown. To resolve such very small amplitudes, the spontaneous activity must have been really low. I think the authors should show recording traces for at least some of the examples shown as they did in their previous paper.

3C and D: It is unclear how the data in the rows were computed. Each row seems to represent the average of several cells (white triangles). Sometimes it seems that the data of two and sometimes of three neurons contribute to each row. Also, 3C includes data from 38 cells and D from 40 cells but both figures have 12 rows. This does not make sense. To avoid this confusion, I suggest to simply show a separate row for each cell, i.e. 38 rows in C and 40 rows in D.

Fig 6 shows that Ntsr1 cells provide columnar input to L3 cells. I thus expected that silencing these cells in Fig 6E would decrease synaptic inputs from L6 to L3 cells. But this is not the case. This is puzzling and an explanation for this is warranted.

In Fig 3A the anterior end of the hippocampus is taken as the center of A1 but in Fig 6A the anterior part of the hippocampus is aligned with the anterior part of A1. Since the correct assignment of A1 boundaries is a crucial issue in how the circuit mapping studies results are interpreted in relationship to the position in tonotopic axis and in interpreting that data were collected in A1, some more information about how AC boundaries were determined is necessary.

Reviewer #1 (Remarks to the Author):

Here Levy et al address an intriguing lateralization in trans-laminar circuitry in auditory cortex that appears to be related to functional differences in sweep selectivity. The question is interesting, and the authors bring an impressive series of experiments to bear on it. The results are mixed, with some compelling findings and some negative results. Overall it is an interesting story, but I do have some concerns.

Figure 1

The heat maps are visually appealing but make it difficult to see the laminar differences. It is certainly not clear that this data has cellular resolution. There is also no quantification, the laminar differences are just asserted. The format and analysis in Figure 7 is much better — the histograms allow a quantitative comparison and the cellular resolution is easily seen from the image. It's unclear why Figs. 1 and 7 use completely different format and analysis, since they are essentially the same type of experiment. Since the two figures are meant to be directly compared they need to be in the same format.

Figure 1 a,b,c: The lateralization for sweeps and vocalizations looks pretty clear but the dominance of superficial layers is less obvious.

Answer: We thank the reviewer for helping us clarify these data. We show examples of the cellular resolution in the grayscale panels in Figure 1A and B. The heat maps and p-value maps show an averaged activity across 3 animals (for each stimulus), which seem to blur the original single-cell data. However, the signal presented is generated from a single-cell resolution dataset: the intensity of each 3D voxel represents the number of detected positive cells in a 190 μ m neighborhood. We think that averaging the density of neurons with a chosen neighborhood size is a clearer way to analyze a complex distribution visually. This creates a synthetic spatial map of the time-averaged neuronal activity viewed from the surface of the brain. To show more clearly the dominance of superficial layers, and to make Figures 1 and 8 directly comparable, we have added plots with a quantitative layer-by-layer analysis (Figure 1D).

Figure 1 d,e,f: Why is the highest FOS activity seen in somatosensory cortex?

For the anterior activation for vocalizations, the same issue arises as the L/R/laminar result above - the results should include quantification, not just "mostly anterior" or "highest activation was largely in superficial."

Answer: These data are from freely moving animals actively exploring their environment. When we performed sound stimulation the animal's environment changed from silence (5 hours of pre-stimulus acclimation) to sound delivery. The change to sound stimulation led to an increase in the animal's exploration of the environment, vigorous whisking, and thus increased activation of the somatosensory cortex. The hippocampus also shows a lot of activity.

To further substantiate the claim of tonotopic activation in anterior portions of the ACx we have also added quantitative cellular density plots (Figure 1H).

Figure 2e,f is interesting and convincing.

Answer: We thank the reviewer for this compliment.

Figure 4

Do 4 a,b and c,d really belong in the same figure?

In 4c there's no quantification at all, just 2 examples and a statement about the octave-to-mm factor. How many recording sites? How did you arrive at this octave-to-mm factor? This could conceivably go in the methods.

Answer: We thank the reviewer for helping us simplify this figure. For clarity, we moved the tonotopy mapping data (Figure 4C) to Supplementary Figure 3. In the main text we state the number of animals and recording sites we used to calculate the octave-to-mm factor (p. 10). These data were collected from 4 animals and 13 recording sites. And in the Methods we now state how we calculated the octave-to-mm factor (p. 26). In every animal we measured best frequency preference using a silicone probe in 2 to 6 locations within the ACx. We then marked with a DiI track locations that were 1-octave and 3-octaves apart. The animals were perfused and the tracks recovered. The octave-to-mm factor we report in the main text is the average across animals of the cortical distances between DiI tracks recovered.

Many aspects of Figure 4 are quite confusing. 4A: What is the Relative tonotopic axis (mm)? What does 0 on this axis correspond to? What do positive and negative directions correspond to? Is it the same as "Hotspot distance from soma (mm)" in panel D, or is that something different? In the histogram inset to panel D, what is the x-axis? In panels A-B, on the absolute tonotopic axis, what does 0 correspond to and what are the positive and negative directions? How do these axes relate to those on the LSPS maps in Figure 3? In panel E is positive to the right and negative to the left?

It would be helpful to include A-P and Hi-Low freq on all of these plots, and to standardize the axis labeling terminology so that all panels can be directly compared (or not if they shouldn't).

Answer: We thank the reviewer for helping us clarify the terminology in this figure. The "mm" typo was introduced by a font change in the figure, so the symbol for micron became an "m". We have made the axis consistent with the label. The relative and absolute tonotopic axis labels refer to the same anatomical nomenclatures used in Figure 3C, D. We have added additional explanation of these labels in the Figure 4 legend. To clarify, the y-axis is the position of the cell along the tonotopic axis where 0 is the tip of the hippocampus, and the x-axis is the location of the cell in the LSPS map grid. For the latter, all cells are centered on the map and represents the x=0 position. The Methods section has the details of these measurements (p. 28, 29), and we added more clarification in the Figure legends.

In the text description of the results for panel 4D, it's not clear what a cluster is and whether that is the same thing as the 3 Anterior, Middle, Posterior groups.

Answer: In the revised manuscript, we now consistently use the term "clusters" not groups to refer to the clusters of Anterior, Middle and Posterior cells analyzed separately.

What does the ANOVA analysis described in conjunction with panel 4D tell us beyond what was

already shown in the linear regressions in panels A, B? Is it a distinct finding or just another way of statistically testing the same question? Explain.

Answer: the ANOVA analysis is a more rigorous approach to tease apart the observations from the linear regressions. We specifically wanted to test the statistical significance of the intra and interhemispheric changes observed qualitatively in the linear regressions. In the revised version we make this point more clearly (p. 10).

Figure 5b

The polar plots are unclear and the data are not convincing of a difference in axonal projection patterns. What does the polar angle correspond to? What does 0 degrees correspond to? What is the polar radius? The fills in 5a make it look like cartesian coordinates of the A-P tonotopic axis and the pia-WM depth axis would be appropriate for analysis, but I can't understand how that relates to the polar plots. Are these cells patched in slices? How do you know the important axons aren't cut off in slicing? A single slicing artifact outlier could greatly skew the average of just 5 cells.

Answer: We thank the reviewer for helping us clarify this data set. Polar angle corresponds to the anatomical orientation shown in Figure 5A. In other words, 0 deg=posterior direction, 90=pia, 180=anterior, 270=white matter. Figure 5 and its legend have been revised to show and state this explicitly. The polar radius is the total normalized, weighted axon/dendrite length density as a function of polar angle. In other words, the polar plots are histograms, so the distance of each data point from the center does not represent anatomical distance but rather the amount of axon/dendrite length within each wedge (c.f. Stepanyants and Chklovskii, 2005). The weighted process length value was calculated by multiplying the length of each axon/dendrite segment by its distance from the soma, to assign greater weight to those processes that extended a significant distance from the soma. Then each plot was normalized to its largest value. The Methods section has been amended to clarify this (p. 32). We have added Cartesian plots of process length density to the supplementary information (Supplementary Figure 5) to clarify the relationship to the polar plots. There is a close relationship between them; i.e., the polar plots are as if one divided the Cartesian plots into pizza-shaped slices centered on the soma, and added up the axon/dendrite length in each slice, weighted by distance as described above. The cells were reconstructed from patched cells in slices. All cells were deeper than 50 microns from the surface of the slice. The Methods section (p.31) has been amended to state this explicitly. We addressed cut axons and outliers by adding Supplementary table 4, reporting the number of intact and severed processes (both axons and dendrites) for each reconstructed cell. The total fraction of cut axons did not differ significantly between the anterior and posterior groups and was reasonably consistent across individual cells. We have also addressed the issue of slicing artifacts in some detail in Levy and Reyes, J Neurosci. 32:5609 (2012). The cell intactness is considerably higher in the present study, most likely because we targeted cells >50 microns deep to the cut surface. We cannot entirely exclude the possibility that slicing introduced some bias particularly in longer-range connectivity, so text has been added to the results (p.13) to address this issue.

Figure 6

When you introduce the synaptic silencer (hM4D-nrxn1a), you should state up front whether it acts pre- or post-synaptically.

Answer: We thank the reviewer for this clarification. The fact that it acts pre-synaptically is now stated clearly in the revised version, p.14.

Why is the positive control in 6c,d analyzed/plotted differently than the negative result in 6e? Either report both in terms of % decrease, or in absolute pA. I assume that the rank-sum tests were performed on the same type of data, rather than the positive control on % change and the negative result on pA. A negative result with a sample size of 5 should include some sort of power analysis to assess the likelihood of a false negative due to small n.

Answer: We thank the reviewer for pointing out the difference in analysis. All chemogenetic data is now analyzed as percent decrease. We also moved the intra and interlaminar L6 result to the supplementary materials (Supplementary Figure 7) given that it is a control and can lead to confusion about the main results. We performed a statistical power analysis for the chemogenetic data, and given the effect sizes measured, our sample size, and an alpha of 0.05 the power was greater than 0.80. We now state this in the Methods (p. 32).

Figure 7.

FYI, the expression is invisible when the figure is printed out.

The axes are not labeled, I am guessing that the numbers refer to cortical depth and distance along the tonotopic axis. Why are the layout and analysis completely different from that in Figure 1? I prefer the format in Figure 7 because it includes some quantification, whereas in Figure 1 the reader can only eyeball the Fos expression. I don't understand why the two figures are formatted completely differently, it prevents a direct comparison between the two, although the main point of Figure 7 is that it's different from Figure 1.

Answer: We thank the reviewer for pointing out the dimness of the image. We have increased the brightness of the picture and, at least in our printer, the expression looks brighter. We also labeled the axes in the plots. As mentioned in our response to the first comment, we have now included cell density plots along the tonotopic and laminar axes in Figure 1 so it can be directly related to the data in Figure 7 (now Figure 8).

Figure 8, p.16 - You should clarify to what extent your results can distinguish between the L6→L3 connection being maturational (develops purely as a function of age) and experience-dependent. It seems to me the noise-rearing experiment shows it is experience dependent, but under normal rearing conditions it isn't clear whether its appearance depends on the onset of sensory input or on maturational age.

Answer: We thank the reviewer for helping us clarify this point. The data shown in Figure 8B suggests that the L6→L3 projection is experience-dependent. Noise-reared animals don't develop the projection whereas age-matched controls do develop it. We have made this point clearer in the Figure by using clearer labels and adding the exact ages compared. We also discuss it in page 21 of the Discussion.

Figure 8 should include the ages on the panels.

Answer: ages are now included in each panel.

Discussion:

The inside-out development of cortical layers should be supported with a citation.

Answer: We have added a reference to support the inside-out development of cortical layers.

You speculate that left ACx could be specialized for detecting specific sequences that are perhaps prevalent in mouse vocalizations. This is a fairly empty statement, but could be greatly strengthened by relating your findings to the growing literature on mouse vocalizations. Which sequences are actually prevalent in mouse vocalizations that left ACx would be specialized for?

Answer: We thank the reviewer for pointing out the functional relevance of our findings. We discuss the possible link between our findings and the statistical features of mouse vocalizations in page 21-22 of the Discussion. We point to Holy and Guo's analyses where it was shown that downward pitch jumps are the most prevalent in male songs. We now emphasize that this is perhaps the functional reason for the prevalence of the high frequency to low frequency (i.e. downward) L6→L3 projection in the left ACx.

It seems like the killer experiment would be to block the L6→L3 projection in right ACx, to see if it abolishes FM sweep selectivity there. Ntsr1 cells didn't work here, so how might you do it? I think this would be a cool future direction to include in the discussion, if you're OK disclosing that.

Answer: This is indeed a cool future direction. We did not go into details because we are developing several approaches and getting into any of them would turn into a long thought experiment and derail the Discussion. There is no easy experiment to isolate the contribution of a neuronal population with unknown genetic markers. We state that we are trying to find these markers in future experiments (p. 21 of the Discussion).

You should discuss how your results fit into the framework laid out by Metherate (for review see Metherate et al. 2005, Hearing Research). There are a whole series of papers there that are highly relevant.

Answer: we agree that this review is very relevant and have added it to the revised manuscript (p. 19). This review by Metherate supports the idea of intracortical horizontal connections being responsible for the very broad receptive fields that have been measured in the auditory cortex. Our study shows that the L6 to L3 projection is one of these horizontal intracortical pathways that can allow integration across multiple frequency bands. We thank the reviewer for bringing this review to our attention.

Reviewer #2 (Remarks to the Author):

Levy et al. demonstrate differences in left and right auditory cortex of mice in respect to responses to sound, layer 6 to layer 3 connectivity, and axonal projections. Furthermore, they show that the emergence of at least some of these these differences depends on normal early acoustic experience. The study is an impressive tour de force incorporating a very large array of techniques and covering a wide variety of circuits aspects. Together the results presented in this

study are a significant step towards understanding the mechanisms that give rise to lateralization of function in auditory cortex.

Specific comments:

Due to the differences in interlaminar connectivity along and across the tonotopic axis (Oviedo et al., 2010), the exact slice orientation relative to the tonotopic axis is a crucial factor that influences the interlaminar connectivity observed in the present scanning and axonal reconstruction experiments. Since slight variations inevitably occur in each preparation, data need to come from a larger number of animals to reduce this influence. This the authors should state the number of animals included for every experiment, not only the number of cells. In order to exclude the possibility that slice orientation is not a contributing factor, I also suggest to analyze whether ‘columnar offset’ of L6 to L3 shows some subject correlations.???

Figure 3 A: The lower panel photo is identical to the photo the senior author has previously published in Nat. Neuroscience 2010. Using the same photo twice is considered self-plagiarisms. The lookup tables change for each example cell, which makes it difficult to compare the maps. I assume peak amplifies in pA are shown. To resolve such very small amplitudes, the spontaneous activity must have been really low. I think the authors should show recording traces for at least some of the examples shown as they did in their previous paper.

Answer: We thank the reviewer for pointing out that we did not include the number of animals in this data set. We now have this information in the main text (p. 9). In the revised version there is a new picture of a slice in panel 3A. We also show representative traces of the synaptic input for each cell in Figure 3B (Supplementary Figure 1). The color scale is in average pA, not peak pA. By design, there is not much spontaneous activity in our experiments: for laser scanning photostimulation one increases the calcium and magnesium concentration (4:4) to lower the probability of spontaneous synaptic events. Therefore even small synaptically driven activity is very easy to isolate because there is virtually no background activity. This is clear to see in the traces included in the Supplementary Figure 1. The Methods section has more details on the recording conditions as well as previous publications on LSPS cited in the manuscript.

Slice orientation has several impacts on circuit mapping. First, in our 2010 study (Oviedo et al Nature Neuro) we reported that orthogonal slices through the ACx (coronal vs horizontal) capture distinct circuit-motifs (see that study for details). Because we found out-of-column motifs only in horizontal slices (that capture tonotopic axis of the ACx) we focused on this particular slice cut in this study. The second impact of slice orientation is the particular angle at which a slice is cut, which directly determines which slice has the best-preserved processes (axons and dendrites). We followed published ACx slicing procedures developed by Cruikshank et al (J Neurophys 2002), where a 15 degree cutting angle from the horizontal plane is used to produce an optimal slice through the ACx. There is only one optimal slice per animal using this procedure, which we determine by examining how parallel the dendrites are to the surface of the slice. When dendrites are parallel to the surface of the slice this indicates that axons are also best preserved. We describe this briefly in page 27 of the Methods, and we have added the Cruikshank et al reference. Regarding subject correlations for “columnar offset”, the circuit mapping data set was collected from 55 animals and every animal contributed 1-3 cells to the data set. Therefore, a few outliers do not underlie the out-of-column input. This is apparent in the population plot of Figure 3C and D, where each row of the 2D matrix represents the

average of 2-3 neighboring cells, as well as Figure 4A-B showing hotspot distance for individual cells.

3C and D: It is unclear how the data in the rows were computed. Each row seems to represent the average of several cells (white triangles). Sometimes it seems that the data of two and sometimes of three neurons contribute to each row. Also, 3C includes data from 38 cells and D from 40 cells but both figures have 12 rows. This does not make sense. To avoid this confusion, I suggest to simply show a separate row for each cell, i.e. 38 rows in C and 40 rows in D.

Answer: We thank the reviewer for helping us clarify these data. The white triangles shown are simply a schematic representation of where the cell bodies are centered on the map, they are not meant to show actual cell numbers per bin. To avoid confusion we have replaced the white triangles with a dashed line. On page 8 and 30 of the manuscript we state that we averaged together the input to cells that were within 50 microns from each other. Therefore, there are approximately 3 cells per row. The 2D plots in Figure 3C and D are meant to capture mesoscale trends not individual cells. We show 6 individual cell examples in Figure 3B. We also show individual cell data in Figure 4 where we analyze hotspot distance to soma. All the data points in Figure 4A and B are from individual cells as stated in p. 9.

Fig 6 shows that Ntsr1 cells provide columnar input to L3 cells. I thus expected that silencing these cells in Fig 6E would decrease synaptic inputs from L6 to L3 cells. But this is not the case. This is puzzling and an explanation for this is warranted.

Answer: We thank the reviewer for this comment and we now provide a clearer explanation of these results in the revised manuscript (p. 14), and a Supplementary Figure (Supp. Fig. 6) comparing the relative strengths of columnar and out-of-column pathways. The CRACM and DREADD experiments involve very different techniques; therefore, to avoid confusion we have now separated these results into two figures. In the CRACM experiments (Figure 6) we *only* activate L6-Nrst1 axons, and we found this projection to be columnar. Only half the cells we mapped had L6-Ntsr1 input at all. We re-plotted the CRACM data as normalized input to control for several experimental variables. First, the amplitudes (but not the organization) of the synaptic currents differed between animals due to viral expression and second, there is likely to be over-representation of input from some channelrhodopsin-positive axons. In the DREADD experiments (Figure 7) we go back to glutamate uncaging in which we're activating all cell types, and use CNO to silence L6-Ntsr1 projections. We found that the out-of-column projection is unchanged. But to directly address the reviewer's question we went back to our DREADD data set and below (Reviewer Figure 1) we show a cell with weak columnar input and stronger out-of-column input. Only the columnar input was blocked by CNO.

Reviewer Figure 1. Colormap shows glutamate uncaging input before CNO application. Bottom traces show (1) out-of-column EPSCs, and (2) columnar EPSCs corresponding to the boxed areas in the colormap.

In Fig 3A the anterior end of the hippocampus is taken as the center of A1 but in Fig 6A the anterior part of the hippocampus is aligned with the anterior part of A1. Since the correct assignment of A1 boundaries is a crucial issue in how the circuit mapping studies results are interpreted in relationship to the position in tonotopic axis and in interpreting that data were collected in A1, some more information about how AC boundaries were determined is necessary. **Answer: We thank the reviewer for helping us clarify this point. Figure 6A is a confocal image of chr2-expressing L6 cells in A1. We showed the approximate location of a cell mapped and the size of the grid. Every map is centered on the cell recorded, but we systematically measure the distance from the cell to the anterior tip of the hippocampus, which we use as our “absolute” tonotopic location. The size of A1 in the mouse is approximately 1.5mm, therefore we erred on the side of being more conservative and restricted mapping to cells that were less than 0.48mm anterior and 0.48mm posterior to the tip of the hippocampus (our x=0 position). We now state this more clearly in the Methods section and provide a citation (p. 29).**

REVIEWERS' COMMENTS:

Reviewer #1 (Remarks to the Author):

The authors have done a great job responding to the comments of both reviewers and the manuscript is much improved. I support publication. I have two minor suggestions that could further strengthen the paper.

1. In the Results referring to Fig. 1A,B,C,D there are a few statements of statistically significant differences (for example: "significantly more Fos positive neurons in the right ACx than the left ACx") that should be supported with a p-value. I realize panels C, G, K show p-value maps but they are binary. Other results in the text are reported with p-value and test name, so it seems odd not to include that for this important result.

2. Figure 3 C,D is easy to mis-read. Reviewer 2 also had comments about this. The dotted line is better but I still got stuck here. I think there are two sources of confusion that the authors could easily correct. First, it's not obvious that each row is L6 from different sets of cells. The title of the figure panel does contain "L6→ L3" but it would really help the reader to mention this in the text and/or figure legend, since it's crucial information for understanding the figure. If you read the legend for Figure 3 it's entirely plausible to assume that the "2D population maps" in panel C are just population-averaged single-cell maps like those in panel B. Second, one reason it's not obvious that each row is from L6 is that the overall visual form of the panel is very similar to those in panel B, even though the data are a quite different form. You should consider some way of presenting panels C,D that makes them look different from panel B, such as a different color scheme, aspect ratio, or other some other layout difference.

Reviewer #2 (Remarks to the Author):

The authors have done a thorough job in addressing and clarifying my critiques and questions and have revised the manuscript accordingly. I have no further comments.

Response to reviewers:

Reviewer #1 (Remarks to the Author):

The authors have done a great job responding to the comments of both reviewers and the manuscript is much improved. I support publication. I have two minor suggestions that could further strengthen the paper.

1. In the Results referring to Fig. 1A,B,C,D there are a few statements of statistically significant differences (for example: "significantly more Fos positive neurons in the right ACx than the left ACx") that should be supported with a p-value. I realize panels C, G, K show p-value maps but they are binary. Other results in the text are reported with p-value and test name, so it seems odd not to include that for this important result.

Response: We thank the reviewer for this suggestion. We now include a significance statement with these data (p. 5).

2. Figure 3 C,D is easy to mis-read. Reviewer 2 also had comments about this. The dotted line is better but I still got stuck here. I think there are two sources of confusion that the authors could easily correct. First, it's not obvious that each row is L6 from different sets of cells. The title of the figure panel does contain "L6→ L3" but it would really help the reader to mention this in the text and/or figure legend, since it's crucial information for understanding the figure. If you read the legend for Figure 3 it's entirely plausible to assume that the "2D population maps" in panel C are just population-averaged single-cell maps like those in panel B. Second, one reason it's not obvious that each row is from L6 is that the overall visual form of the panel is very similar to those in panel B, even though the data are a quite different form. You should consider some way of presenting panels C,D that makes them look different from panel B, such as a different color scheme, aspect ratio, or other some other layout difference.

Response: We thank the reviewer for these suggestions to improve the clarity of this figure. We changed the colormaps in Figure 3D and E to reflect the different data being plotted. And we also added text in the figure legend to explain the 2D population map binning.

Reviewer #2 (Remarks to the Author):

The authors have done a thorough job in addressing and clarifying my critiques and questions and have revised the manuscript accordingly. I have no further comments.

Response: We thank the reviewer for their help in improving the manuscript.